# S$^2$NN: Sub-bit Spiking Neural Networks

**Wenjie Wei**[1], **Malu Zhang**[1,2]*, **Jieyuan Zhang**[1], **Ammar Belatreche**[3], **Shuai Wang**[1],
**Yimeng Shan**[1,4], **Hanwen Liu**[1], **Honglin Cao**[1], **Guoqing Wang**[1], **Yang Yang**[1], **Haizhou Li**[2,5]

[1]University of Electronic Science and Technology of China,
[2]Shenzhen Loop Area Institute, [3]Northumbria University, [4]Liaoning Technical University
[5]The Chinese University of Hong Kong, Shenzhen (CUHK-Shenzhen)

## Abstract

Spiking Neural Networks (SNNs) offer an energy-efficient paradigm for machine intelligence, but their continued scaling poses challenges for resource-limited deployment. Despite recent advances in binary SNNs, the storage and computational demands remain substantial for large-scale networks. To further explore the compression and acceleration potential of SNNs, we propose Sub-bit Spiking Neural Networks (S$^2$NNs) that represent weights with less than one bit. Specifically, we first establish an S$^2$NN baseline by leveraging the clustering patterns of kernels in well-trained binary SNNs. This baseline is highly efficient but suffers from *outlier-induced codeword selection bias* during training. To mitigate this issue, we propose an *outlier-aware sub-bit weight quantization* (OS-Quant) method, which optimizes codeword selection by identifying and adaptively scaling outliers. Furthermore, we propose a *membrane potential-based feature distillation* (MPFD) method, improving the performance of highly compressed S$^2$NN via more precise guidance from a teacher model. Extensive results on vision tasks reveal that S$^2$NN outperforms existing quantized SNNs in both performance and efficiency, making it promising for edge computing applications.

## 1 Introduction

Spiking Neural Networks (SNNs), with their unique event-driven paradigm, are seen as a promising energy-efficient solution for realizing the next generation of machine intelligence [1, 2]. Specifically, SNNs employ binary spikes for information transmission and process them in a sparse event-driven manner. This computational paradigm transforms convolution operations in traditional artificial neural networks (ANNs) from computationally intensive multiply-accumulate (MAC) to efficient accumulate (AC), thereby significantly improving computational efficiency [3]. Moreover, the event-driven nature of SNNs has spurred the development of neuromorphic hardware, such as SpiNNaker [4], TrueNorth [5], Loihi [6], and Tianjic [7], further harnessing their potential for energy efficiency. However, as large language models (LLMs) exhibit superior performance, the SNN community has begun scaling up SNN models to improve their performance on complex tasks [8, 9, 10]. While this scaling has enhanced performance, it has sacrificed SNNs' inherent efficiency advantages, posing storage and computational challenges for their deployment on resource-constrained edge devices.

In recent years, researchers have increasingly investigated compression techniques for SNNs, such as pruning [11, 12], neural architecture search [13, 14], quantization [15, 16], and others [17, 18, 19]. As an extreme quantization technique, binarization restricts parameters to only two values, i.e., -1 and +1 [20]. By applying binarization, researchers have developed lightweight binary SNNs (BSNNs). BSNNs not only inherit the sparse event-driven paradigm of SNNs, but also further convert

---

*Corresponding author: maluzhang@uestc.edu.cn

convolution operations from AC to cost-effective bitwise operations. This greatly reduces the resource overhead of SNNs, especially on edge devices. However, as neural networks are scaled to greater depths to meet practical demands, the computational burden remains a significant challenge, even for binary-weighted versions [21, 22]. This raises an important question: "Can the compression and acceleration potential of SNNs be further exploited?"

Studies on Binary Neural Networks (BNNs) have shown that binarized convolutional kernels in well-trained BNNs exhibit clustering patterns within each layer. This phenomenon becomes increasingly pronounced as the network depth increases [23, 24]. In the case of a 3×3 kernel, an analysis of the distribution of all possible 3×3 kernel values ($2^{3\times3}$, representing the full codebook) reveals that only a small subset of binary kernels (codewords) is frequently activated. Based on this observation, kernels in each layer can be restricted to a subset of binary convolution kernels (a compact codebook) during training, enabling sub-bit model compression. This method achieves higher compression ratios and faster inference speeds compared to BNNs. However, despite these promising efficiency gains, sub-bit techniques remain unexplored in the context of SNNs.

In this paper, we introduce sub-bit spiking neural networks ($S^2$NNs) to further harness the compression and acceleration potential of SNNs. We first construct an $S^2$NN baseline that encodes weights using less than 1 bit. However, we observe that this baseline is prone to outliers when mapping 32-bit kernels to a binary kernel subset, resulting in suboptimal binary kernel selection. To address this issue, we propose an underline{o}utlier-aware underline{s}ub-bit weight quantization (OS-Quant) method that improves binary kernel selection by identifying and scaling outliers. Furthermore, to enhance the baseline performance, we introduce a underline{m}embrane potential-based underline{f}eature underline{d}istillation (MPFD) method, which utilizes a teacher model to guide the training of the highly compressed baseline. The main contributions are as follows:

- We introduce a $S^2$NN baseline that achieves extreme model compression by encoding weights with less than 1 bit. This approach achieves higher compression ratios than BSNNs, further unlocking the potential of SNNs in terms of both compression and acceleration.

- We identify that the baseline suffers from outlier-induced codeword selection bias, negatively impacting performance. To address it, we propose an outlier-aware sub-bit weight quantization (OS-Quant). OS-Quant effectively eliminates the influence of outliers on quantization while preserving the spatial features of kernels, ensuring optimal codeword selection.

- We propose a membrane potential-based feature distillation (MPFD) framework which employs a teacher model to guide the training of the highly compressed baseline. By applying distillation at the membrane potential level, MPFD achieves more accurate knowledge transfer, improving performance without compromising compression benefits.

- Extensive experiments demonstrate that integrating OS-Quant and MPFD into the baseline enables $S^2$NN to achieve state-of-the-art (SOTA) performance and efficiency. Furthermore, tests across diverse tasks and architectures validate the scalability of our method.

## 2 Related Works

**Binary Neural Network**    Binarization is traditionally considered the most extreme quantization method, which helps reduce computational overheads but compromises model accuracy. Therefore, most early BNN research focuses on narrowing the gap between BNNs and full-precision models. For example, [25] propose floating-point scaling factors for BNNs to fit real-value weights. [26] approximate real-value weights by linearly combining multiple binary weight bases. [27] propose adding shortcuts similar to ResNet to reduce information loss during the binarization process. [28] retains information in BNNs by maximizing information entropy and minimizing gradient errors. [29] adopt a generalized activation function to capture the distribution reshape and shift, achieving excellent accuracy on ImageNet-1K.

As the performance gap between BNNs and full-precision models continues to narrow, a few recent studies have begun to further compress BNNs, successfully reducing parameter bitwidths to less than 1 bit. [30] propose a flexible encryption algorithm that encrypts subvectors of flattened weights into low-dimensional binary codes. [23] observe the kernel clustering distribution characteristic of BNNs and then constrain kernels within a prescribed binary kernel subset during training. [31] applies the concept of stacked low-dimensional filters and product quantization to achieve sub-bit model compression. [32] propose minimum spanning tree compression, which uses the fact that output

channels in binary convolution can be calculated using another output channel and XNOR operations. Recently, [33] has been learning sequences of binary tiles to populate the layers of DNNs, achieving sub-bit storage of neural network parameters.

**Binary Spiking Neural Network**   Given SNNs' binary spike activations, researchers have developed BSNNs with 1-bit synaptic weights to reduce storage and accelerate computation. Early works explore ANN-SNN conversion to obtain BSNNs. For instance, [34] first train a binary convolutional neural network and then convert it to a BSNN. [35] introduces a weight-threshold balanced conversion approach to minimize conversion errors and enhance BSNN performance. However, these conversion-based methods inevitably suffer from accuracy degradation and fail to process sequential datasets. This leads researchers to explore direct BSNN training methods. [36] directly train BSNNs using surrogate gradient (SG) methods. [37] propose a novel Bayesian-based BSNNs learning algorithm that outperforms SG methods in accuracy. [38] presents a time-encoded BSNN where neurons emit at most one spike and learn in an event-driven manner, thereby offering substantial energy benefits. By combining BNN and SNN advantages, [16] propose BitSNN, which enhances energy efficiency through binary weights, single-step inference, and sparse activations. Recently, [15] draw from information theory and introduce a weight-spike dual regulation method, aimed at achieving the performance of full-precision SNNs (FP SNNs) by improving BSNN's information capacity. Despite these advances, current methods still face key limitations. Firstly, these studies mainly focus on narrowing the performance gap between BSNNs and FP-SNNs. However, as networks scale up to meet practical application demands, the computational burden remains a challenge even with binary versions. Secondly, these studies are limited to simple image classification tasks, leaving their scalability to complex tasks and diverse architectures unexplored.

To address these limitations, we propose a novel method to further explore SNNs' potential in compression and acceleration. Moreover, our method emphasizes scalability across complex tasks and diverse architectures, making it practical for broad applications. These efforts will facilitate the efficient deployment of large-scale SNNs on edge devices.

## 3   Sub-bit Spiking Neural Networks Baseline

In this section, we construct a sub-bit SNN baseline by leveraging existing knowledge, primarily including the employed spiking neuron models and the sub-bit weight quantization.

**Spiking Neuron Model**   Various neuron models are proposed to replicate the information processing capabilities of biological neurons, such as Hodgkin-Huxley [39], Izhikevich [2], and Leaky Integrate-and-Fire (LIF) [40] models. Due to its computational efficiency, the LIF model is widely used. Therefore, we also employ the classic LIF model in our work; its membrane potential is described as,

$$\tilde{\mathbf{u}}^\ell[t] = \tau \mathbf{u}^\ell[t-1] + f(\mathbf{w}_f^\ell, \mathbf{s}^{\ell-1}[t]), \tag{1}$$

$$\mathbf{s}^\ell[t] = \Theta(\tilde{\mathbf{u}}^\ell[t] - \theta), \tag{2}$$

$$\mathbf{u}^\ell[t] = \tilde{\mathbf{u}}^\ell[t] \left(1 - \mathbf{s}^\ell[t]\right), \tag{3}$$

where $\tau$ is the leaky factor, $\mathbf{w}_f^\ell$ is the 32-bit weight matrix of layer $\ell$, $f(\cdot)$ is the convolution or linear operation followed by batch normalization (BN), and $\Theta(\cdot)$ is the Heaviside step function. As described above, neurons integrate inputs and emit a spike $\mathbf{s}$ when the membrane potential $\tilde{\mathbf{u}}$ exceeds the threshold $\theta$. After each spike emission, the hard reset mechanism is invoked, where $\mathbf{u}$ is reset to zero upon emitting a spike and remains unchanged in the absence of a spike.

**Sub-Bit Weight Quantization**   The $S^2$NN baseline is constructed based on the sub-bit weight quantization technique, as shown in Fig. 1(b). This technique is applied to the binary convolutional kernels of the network. Therefore, we first construct a BSNN by quantizing the weight matrix $\mathbf{w}_f^\ell$ in Eq. (1) to a 1-bit representation, described as,

$$\mathbf{w}_b^\ell = \alpha \cdot \text{sign}(\mathbf{w}_f^\ell), \ \text{sign}(\mathbf{w}_f^\ell) = \begin{cases} -1, & \text{if } \mathbf{w}_f^\ell < 0, \\ +1, & \text{otherwise,} \end{cases} \tag{4}$$

where $\alpha$ is the channel-wise scaling factor that is calculated as the average of the absolute value of weights in each output channel [25], and $\mathbf{w}_b^\ell$ is the binary weight matrix. By combining this

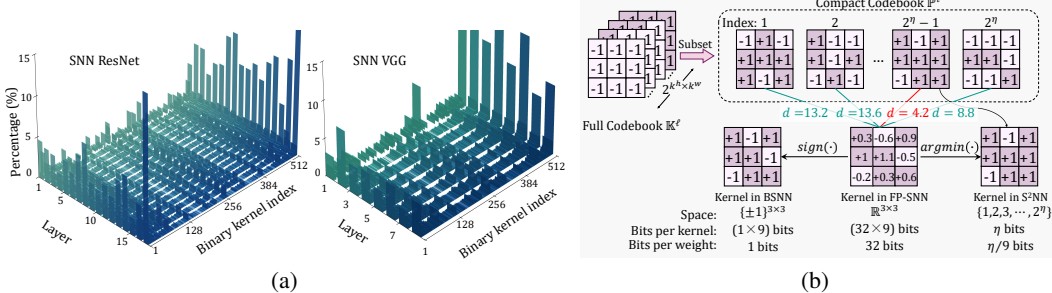

(a)               (b)

Figure 1: (a) Convolutional kernels in well-trained BSNNs exhibit clustering patterns. This motivates us to achieve higher compression ratios than BSNNs by using a compact codebook $\mathbb{P}$ instead of the full codebook $\mathbb{K}$. (b) The constructed S$^2$NN baseline.

binarization with Eq. 1, the convolution or linear operation can be transformed from $\mathbf{w}_f^\ell \circledast \mathbf{s}^{\ell-1}[t]$ to $\alpha \cdot (\mathrm{sign}(\mathbf{w}_f^\ell) \oplus \mathbf{s}^{\ell-1}[t])$, where arithmetic operations are replaced with efficient bitwise operations $\oplus$. For simplicity, we omit the scaling factor in subsequent analysis as it is applied after the bitwise convolution operation.

Before introducing sub-bit weight quantization, we first present the clustering pattern of binary kernels. We formulate the weight matrix $\mathbf{w}_b^\ell$ as the channel-wise concatenation of each binary kernel, i.e., $\mathbf{w}_b^\ell = \big\|_{c=1}^{c_{out}^\ell \cdot c_{in}^\ell} \mathbf{w}_{b,c}^\ell$, where $c_{out}^\ell$ and $c_{in}^\ell$ are the number of output and input channels in layer $\ell$, respectively. Each binary kernel $\mathbf{w}_{b,c}^\ell \in \mathbb{K}$ is derived by $\mathbf{w}_{b,c}^\ell = \mathrm{sign}(\mathbf{w}_{f,c}^\ell)$. Here, $\mathbb{K} = \{\pm 1\}^{k_w \cdot k_h}$ denotes the set of all possible binary kernels (i.e., **full codebook**) with size $k_w \times k_h$. This full codebook contains $|\mathbb{K}| = 2^{k_w \cdot k_h}$ unique binary kernels (i.e., **codewords**). Previous studies reveal these codewords exhibit layer-dependent clustering patterns in well-trained BNNs, especially in deep layers [23, 24]. We conduct a similar analysis in well-trained BSNNs and observe the same phenomenon, as illustrated in Fig 1(a). Based on prior studies and the clustering patterns of BSNNs, we use a compact codebook rather than the full codebook $\mathbb{K}$ to construct the S$^2$NN baseline. We present in Appendix A the top-k codeword proportions for BSNN. The findings validate the clustering and reveal increased clustering patterns in deeper layers.

The S$^2$NN baseline is built by (1) sampling layer-specific codeword subsets $\mathbb{P}^\ell$ (i.e., **compact codebook**), (2) mapping each 32-bit kernel to its nearest codeword in $\mathbb{P}^\ell$ for inference, depicted in Fig. 1(b). It is defined as [23],

$$\text{Forward propagation: } \mathbf{w}_{b,c}^\ell = \arg\min_{\mathbf{k} \in \mathbb{P}^\ell} \big\| \mathbf{k} - \mathbf{w}_{f,c}^\ell \big\|_2^2, \tag{5}$$

$$\text{Backward propagation: } \frac{\partial \mathcal{L}}{\partial \mathbf{w}_{f,c}^\ell} = 1_{|\mathbf{w}_{b,c}^\ell| \le 1} \cdot \frac{\partial \mathcal{L}}{\partial \mathbf{w}_{b,c}^\ell}, \tag{6}$$

where $\mathbb{P}^\ell \subset \mathbb{K}$, $|\mathbb{P}^\ell| = 2^\eta$, and $\eta < k_w \cdot k_h$. Noteworthy, each binary codeword in $\mathbb{P}^\ell$ can be optimized during training, so $\mathrm{sign}(\cdot)$ must be applied after optimization to preserve its binary representation. By integrating Eq. (5) with Eq. (1), the S$^2$NN baseline is established.

The sub-bit quantization in Eq. (5) computes the squared L2 distance between $\mathbf{w}_{f,c}^\ell$ and each candidate codeword $\mathbf{k}$ in $\mathbb{P}^\ell$, and selects the nearest $\mathbf{k}$ to replace $\mathbf{w}_{f,c}^\ell$ for forward propagation. This approach achieves below 1-bit compression by using an index to represent each binary kernel. Specifically, for weights in the $\ell$-th layer, i.e., $\mathbf{w}_b^\ell \in \{\pm 1\}^{c_{in}^\ell \cdot c_{out}^\ell \cdot k_w \cdot k_h}$, BSNN requires $k_w \cdot k_h \cdot c_{in}^\ell \cdot c_{out}^\ell$ bits to store the whole parameters, while the S$^2$NN baseline requires bits of $\eta \cdot c_{in}^\ell \cdot c_{out}^\ell$ for indicies and $2^\eta \cdot k_w \cdot k_h$ for the storage of $\mathbb{P}^\ell$. Since $\eta$ is designed to be smaller than $k_w \cdot k_h$, the S$^2$NN baseline achieves a compression ratio of $\frac{\eta \cdot c_{in}^\ell \cdot c_{out}^\ell + 2^\eta \cdot k_w \cdot k_h}{k_w \cdot k_h \cdot c_{in}^\ell \cdot c_{out}^\ell} \approx \frac{\eta}{k_w \cdot k_h}$ for each weight. Consider the commonly used 3×3 convolutional kernel, the S$^2$NN baseline represents each parameter with 0.44, 0.56, and 0.67 bit when $\eta$ of 4, 5, and 6, respectively. A comprehensive analysis of the baseline's compression and acceleration advantages is presented in Appendix H.

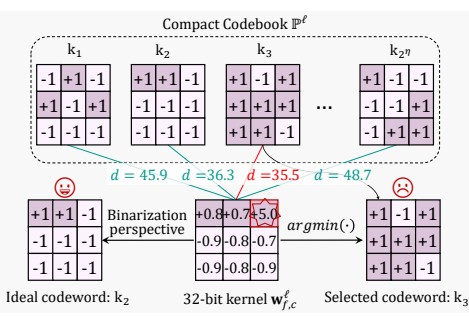 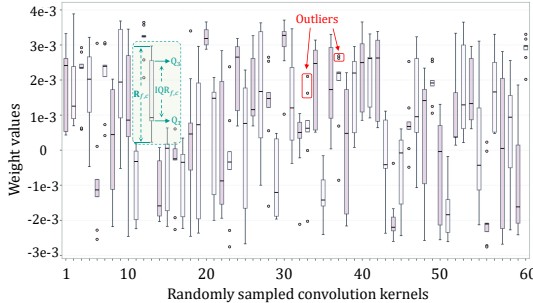

(a) Outlier-induced codeword selection bias  (b) Outliers during the training of the S$^2$NN baseline

Figure 2: (a) The outliers dominate the distance calculations, diminishing the contributions of other elements and leading the baseline to select an undesirable codeword for inference. (b) During the training process of the S$^2$NN baseline, we randomly sample several kernels and analyze their weight distributions using a box plot. Discrete points in the figure indicate outliers.

## 4 Method

In this section, we first analyze how the S$^2$NN baseline is adversely affected by outlier-induced codeword selection bias. Then, we propose an outlier-aware sub-bit weight quantization that eliminates outlier interference while preserving kernels' spatial features. Finally, we introduce a membrane potential-based distillation, improving the S$^2$NN's performance via the guidance of a teacher model.

### 4.1 Outlier-induced Codeword Selection Bias

A key step in the S$^2$NN baseline is to compute the squared L2 distance between the 32-bit kernel $\mathbf{w}_{f,c}^\ell$ and each candidate codeword $\mathbf{k}$ in $\mathbb{P}^\ell$, then select the nearest codeword to replace $\mathbf{w}_{f,c}^\ell$ for inference. While it achieves sub-bit weight compression, this process is susceptible to outliers, leading to biased codeword selection. This bias can be regarded as quantization errors in parameter compression.

To illustrate this issue, we consider an example of a 3×3 kernel, as shown in Fig. 2(a). Given $\mathbf{w}_{f,c}^\ell$ and some candidate codewords. From a binarization perspective, $\mathbf{k}_2$ is the optimal choice to replace $\mathbf{w}_{f,c}^\ell$, as it better maintains the sign patterns of the majority elements in $\mathbf{w}_{f,c}^\ell$. However, the presence of an outlier 5 causes the baseline to choose $\mathbf{k}_3$ instead. We define this inconsistency as the outlier-induced codeword selection bias. This issue occurs since the employed squared L2 distance is sensitive to large values, causing outliers to dominate the distance computation and overshadow the contribution of other elements. Unfortunately, the chosen non-optimal codeword cannot capture the true sign pattern of $\mathbf{w}_{f,c}^\ell$, adversely affecting the baseline learning. In Fig. 2(b), we show that many outliers exist in the learning process, indicating that the baseline suffers from a severe codeword selection bias. This motivates us to address the bias for stable convergence and improved performance. In Appendix B, we count the percentage of kernels containing outliers in each layer to demonstrate that this bias is a common phenomenon.

### 4.2 Outlier-Aware Sub-Bit Weight Quantization

We introduce the OS-Quant to address the codeword selection bias caused by outliers. The OS-Quant comprises two steps: (1) interquartile range (IQR)-based outlier detection, and (2) spatially-aware outlier scaling. This approach effectively mitigates the negative impact of outliers on the quantization process, while simultaneously preserving the spatial information inherent in float-point kernels.

**IQR-based Outlier Detection**  During the sub-bit quantization, we use quartile statistics to determine the boundaries for normal weight values in $\mathbf{w}_{f,c}^\ell$, and consider values outside these boundaries as outliers. Specifically, we first calculate the interquartile range of $\mathbf{w}_{f,c}^\ell$, as defined in [41],

$$\mathrm{IQR}_{f,c}^\ell = \mathrm{Q}_3 - \mathrm{Q}_1, \tag{7}$$

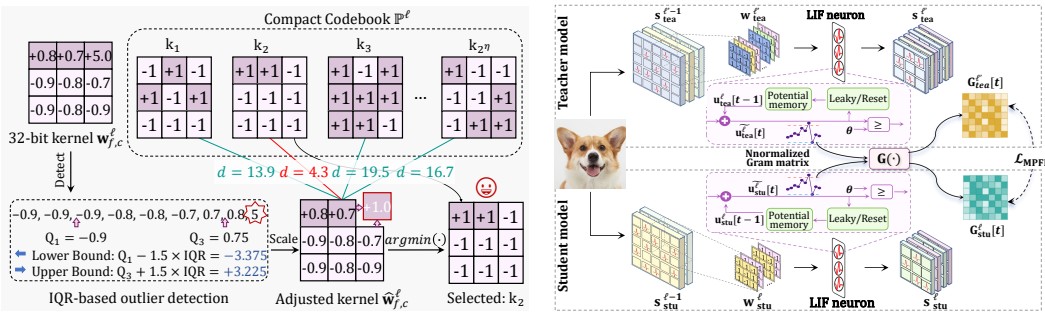

(a) Outlier-aware sub-bit weight quantization      (b) Membrane potential-based feature distillation

Figure 3: Schematic diagram of the proposed OS-Quant and MPFD method.

where $Q_1$ and $Q_3$ denote the first and third quartiles of $\mathbf{w}_{f,c}^{\ell}$, respectively. Then, a threshold coefficient $\gamma$ is applied to define the normal weight range as follows,

$$\mathbf{R}_{f,c}^{\ell} = [Q_1 - \gamma \cdot \mathrm{IQR}_{f,c}^{\ell}, Q_3 + \gamma \cdot \mathrm{IQR}_{f,c}^{\ell}], \tag{8}$$

where $\gamma$ is a coefficient that controls the sensitivity of outlier detection. A larger $\gamma$ yields fewer outliers, while a smaller $\gamma$ leads to more. Following the well-established 'Tukey's fences' [42], $\gamma$ is typically set to 1.5. Detail analysis of this hyperparameter is provided in Appendix C. Accordingly, any weights outside this range are classified as outliers. We define the set of outlier coordinates in $\mathbf{w}_{f,c}^{\ell}$ as,

$$\mathbf{O}_{f,c}^{\ell} = \left\{ (i,j) \mid \mathbf{w}_{f,c}^{\ell}(i,j) \notin \mathbf{R}_{f,c}^{\ell} \right\}, \tag{9}$$

where $(i,j)$ is the coordinates in the kernel, and $\mathbf{w}_{f,c}^{\ell}(i,j)$ corresponds to the weight value at the specified position. As shown in Eq. (7), the interquartile range metric focuses on the central 50% of all weights in the kernel, thus being less affected by outliers. This makes our outlier detection method exhibit greater robustness than methods relying on mean and variance, thereby providing a more reliable kernel adjustment to resolve the codeword selection bias.

**Spatially-Aware Outlier Scaling**   After detecting outliers, we propose a spatially-aware scaling approach to eliminate their impact on distance computation. This method leverages the relationships between outliers and their spatial neighbors, preserving the spatial feature of 32-bit kernels.

Given the outlier set $\mathbf{O}_{f,c}^{\ell}$ of the kernel $\mathbf{w}_{f,c}^{\ell}$, we determine spatial neighbors for each outlier within this set. For an outlier located at $(i,j) \in \mathbf{O}_{f,c}^{\ell}$, its neighbor set is defined as,

$$\mathbf{N}_{(i,j)} = \{(i \pm 1, j), (i, j \pm 1)\} \cap [1, k_w] \times [1, k_h], \tag{10}$$

where $k_w$ and $k_h$ denote the kernel's width and height, respectively. This set effectively represents the local spatial relationships of the outlier within the kernel. Based on this spatial information, we introduce a regularization term to adaptively scale the outliers, and it is computed as,

$$\Omega_{i,j} = \frac{1}{|\mathbf{N}_{(i,j)}|} \sum_{(p,q) \in \mathbf{N}_{(i,j)}} \left| \mathbf{w}_{f,c}^{\ell}(i,j) - \mathbf{w}_{f,c}^{\ell}(p,q) \right|, \tag{11}$$

where $|\mathbf{N}_{(i,j)}|$ is the number of spatial neighbors. $\Omega_{i,j}$ is calculated as the mean of the absolute differences between an outlier and its neighbors. Then, we derive an adjusted 32-bit kernel $\hat{\mathbf{w}}_{f,c}^{\ell}$,

$$\hat{\mathbf{w}}_{f,c}^{\ell}(i,j) = \begin{cases} (1/\Omega_{i,j}) \cdot \mathbf{w}_{f,c}^{\ell}(i,j), & \text{if } (i,j) \in \mathbf{O}_{f,c}^{\ell}, \\ \mathbf{w}_{f,c}^{\ell}(i,j), & \text{otherwise.} \end{cases} \tag{12}$$

As a result, the OS-Quant method is described as follows,

$$\text{Forward: } \mathbf{w}_{b,c}^{\ell} = \arg\min_{\mathbf{k} \in \mathbb{P}^{\ell}} \left\| \mathbf{k} - \hat{\mathbf{w}}_{f,c}^{\ell} \right\|^2; \quad \text{Backward: } \frac{\partial \mathcal{L}}{\partial \mathbf{w}_{f,c}^{\ell}} = 1_{|\mathbf{w}_{b,c}^{\ell}| \leq 1} \cdot \frac{\partial \mathcal{L}}{\partial \hat{\mathbf{w}}_{b,c}^{\ell}} \cdot \frac{\partial \hat{\mathbf{w}}_{f,c}^{\ell}}{\partial \mathbf{w}_{f,c}^{\ell}}. \tag{13}$$

In summary, OS-Quant effectively addresses the codeword selection in sub-bit quantization bias by detecting and scaling outliers. Furthermore, in the process of outlier scaling, OS-Quant preserves the spatial features of full-precision kernels, enhancing the training stability and performance of $S^2$NN. We also compare our OS-Quant with several alternative outlier-handling methods in Appendix D.

### 4.3 Membrane Potential-based Feature Distillation

The S$^2$NN baseline yields great efficiency gains but suffers from performance degradation. To overcome this, we introduce a distillation technique to preserve the compressed model's performance [43]. Distillation is categorized into logit-based knowledge distillation (LGKD) and feature-based distillation (FD) [44]. FD typically distills a better-performing student since mimicking the teacher's intermediate features provides the student with more precise optimization directions [45, 46].

In the SNN domain, existing FD methods regard the firing rate of neurons as network intermediate features and aim at aligning the firing rates between teacher and student networks. These firing rate-based FD (FRFD) methods achieve this alignment by adjusting membrane potentials and further controlling the spike generation in backpropagation [47]. Mathematically, the gradient of the distillation loss to the membrane potential is expressed as:

$$\frac{\partial \mathcal{L}_{distill}}{\partial \tilde{\mathbf{u}}^\ell[t]} = \frac{\partial \mathcal{L}_{\mathrm{FRFD}}}{\partial \mathbf{s}^\ell[t]} \cdot \frac{\partial \mathbf{s}^\ell[t]}{\partial g(\cdot)} \cdot \frac{\partial g(\cdot)}{\partial \tilde{\mathbf{u}}^\ell[t]}, \tag{14}$$

where $\frac{\partial \mathcal{L}_{\mathrm{FRFD}}}{\partial \mathbf{s}^\ell[t]}$ can be directly calculated from the distillation loss function. Notably, this distillation-related gradient computation involves an extra surrogate gradient function $g(\cdot)$, which causes the gradient induced by distillation on the membrane potential to be imprecise, thereby compromising the distillation optimization process. As a result, we propose a direct MPFD method at the membrane potential level to achieve more precise optimization directions. Mathematically, within MPFD, the gradient of the distillation loss to the membrane potential $\frac{\partial \mathcal{L}_{distill}}{\partial \tilde{\mathbf{u}}^\ell[t]}$ can be derived directly from the distillation loss function $\mathcal{L}_{\mathrm{MPFD}}$. The MPFD is formulated as,

$$\mathcal{L}_{\mathrm{MPFD}} = \sum_{\{\ell', \ell\} \in \mathcal{P}} \sum_t \left\| \mathbf{G}_{\mathrm{tea}}^{\ell'}[t] - \mathbf{G}_{\mathrm{stu}}^\ell[t] \right\|_2, \tag{15}$$

$$\mathbf{G}_{\mathcal{M}}^\ell[t] = \frac{\mathcal{Q}(\tilde{\mathbf{u}}_{\mathcal{M}}^\ell[t]) \cdot \mathcal{Q}(\tilde{\mathbf{u}}_{\mathcal{M}}^\ell[t])^\mathsf{T}}{\left\| \mathcal{Q}(\tilde{\mathbf{u}}_{\mathcal{M}}^\ell[t]) \cdot \mathcal{Q}(\tilde{\mathbf{u}}_{\mathcal{M}}^\ell[t])^\mathsf{T} \right\|_2}, \tag{16}$$

where $\{\ell', \ell\} \in \mathcal{P}$ denotes layer pairs between teacher and student, $\mathcal{M} \in \{\mathrm{tea}, \mathrm{stu}\}$ is the network type, and $\mathcal{Q} : \mathbb{R}^{b \cdot c \cdot h \cdot w} \to \mathbb{R}^{b \cdot chw}$ is a operation transforming tensor dimensions from $[b, c, h, w]$ to $[b, c \times h \times w]$. In Eq. (16), we introduce a metric $\mathbf{G}$ using a normalized Gram matrix of membrane potentials, which can effectively represent the network's semantic information [48].

We summarize the advantages of MPFD in two aspects. First, by directly imposing distillation on membrane potentials, it achieves more precise knowledge transfer from teacher to student networks. Second, the inner product-based formulation of $\mathbf{G}$ facilitates cross-architecture distillation, without requiring matched network layers or identical layer dimensions between teacher and student networks. As a result, the MPFD significantly enhances the effectiveness and flexibility of knowledge distillation. Further analysis of MPFD is available in Appendix E.

### 4.4 Workflow and Supplementary Details for S$^2$NN

We develop the S$^2$NN by integrating the OS-Quant and MPFD into the baseline, with its workflow described in Algorithm 1. We provide more details of S$^2$NN in the Appendix. In Appendix H, we provide an in-depth discussion of how S$^2$NN achieves below-1-bit compression and its acceleration advantages. In Appendix I, we present a comparison between S$^2$NN and BSNN on an FPGA, highlighting the advantages of S$^2$NN in terms of both compression and acceleration.

The proposed S$^2$NN aims to optimize SNN deployment efficiency. Notably, the involved distance calculations, outlier detection, outlier scaling, and distillation in our methods introduce no additional overhead during inference. After training, only the compact codebook $\mathbb{P}$ and the weight-to-codeword indices are stored. During inference, the model reconstructs binary kernels directly from the indices and $\mathbb{P}$, without performing distance calculations, OS-Quant, or membrane potential–based distillation. Consequently, our S$^2$NN achieves below 1-bit model compression, which further explores the potential of SNNs for both compression and acceleration while maintaining high performance, making S$^2$NN particularly suitable for applications in resource-limited scenarios that require reliable and efficient processing.

**Algorithm 1** One training iteration process of the S$^2$NN.

1: **Input:** Initial SNN model: $\mathcal{M} = \{\mathbf{w}_f^1, \cdots, \mathbf{w}_f^L\}$; Size of $\mathbb{P}$: $\eta$; A well-trained teacher model: $\mathcal{M}_{\text{tea}}$; Input;

2: **Initialize:** Randomly sample layer-specific $\mathbb{P}^\ell$, where $|\mathbb{P}| = 2^\eta$ and $\eta < k_w \cdot k_h$; Initialize an empty list $\mathcal{F}$;

3: **for** $\ell \leftarrow 1$ **to** $L$ **do**

4:    **for** $c \leftarrow 1$ **to** $c_{out}^\ell \cdot c_{in}^\ell$ **do**

5:       $\triangleright$ Calculate IQR to confine a normal weight range: $\mathbf{R}_{f,c}^\ell = [\mathrm{Q}_1 - \gamma \cdot \mathrm{IQR}_{f,c}^\ell; \mathrm{Q}_3 + \gamma \cdot \mathrm{IQR}_{f,c}^\ell]$;

6:       $\triangleright$ Get the coordinates of outliers in the kernel:
$$\mathbf{O}_{f,c}^\ell = \{(\mathrm{i},\mathrm{j}) \mid \mathbf{w}_{f,c}^\ell(\mathrm{i},\mathrm{j}) \notin \mathbf{R}_{f,c}^\ell\};$$

7:       $\triangleright$ Calculate a regularization term for each outlier:
$$\Omega_{\mathrm{i},\mathrm{j}} = \frac{1}{|\mathbf{N}_{(\mathrm{i},\mathrm{j})}|} \sum\nolimits_{(\mathrm{p},\mathrm{q}) \in \mathcal{N}_{(\mathrm{i},\mathrm{j})}} |\mathbf{w}_{f,c}^\ell(\mathrm{i},\mathrm{j}) - \mathbf{w}_{f,c}^\ell(\mathrm{p},\mathrm{q})|,$$

8:       $\triangleright$ Spatially-aware scale each outlier:
$$\hat{\mathbf{w}}_{f,c}^\ell(\mathrm{i},\mathrm{j}) = \begin{cases} (1/\Omega_{\mathrm{i},\mathrm{j}}) \cdot \mathbf{w}_{f,c}^\ell(\mathrm{i},\mathrm{j}), & \text{if } (\mathrm{i},\mathrm{j}) \in \mathbf{O}_{f,c}^\ell, \\ \mathbf{w}_{f,c}^\ell(\mathrm{i},\mathrm{j}), & \text{otherwise}; \end{cases}$$

9:       $\triangleright$ Apply OS-Quant to achieve sub-bit quantization: $\mathbf{w}_{b,c}^\ell = \arg\min_{\mathbf{k} \in \mathbb{P}^\ell} \|\mathbf{k} - \hat{\mathbf{w}}_{f,c}^\ell\|^2$;

10:    **end for**

11:    $\triangleright$ Concatenate and reshape: $\mathbf{w}_b^\ell \leftarrow \text{concat\&reshape}(\mathbf{w}_{b,1}^\ell, \cdots, \mathbf{w}_{b,c_{out}^\ell \cdot c_{in}^\ell}^\ell)$;

12:    $\triangleright$ $\tilde{\mathbf{u}}^\ell[t] = \tau \mathbf{u}^\ell[t-1] + BN(\alpha \cdot (\mathbf{w}_b^\ell \oplus \mathbf{s}^{\ell-1}[t]))$,

13:    $\triangleright$ Record $\tilde{\mathbf{u}}$ for distillation: $\mathcal{F} \leftarrow \mathcal{F}.append(\tilde{\mathbf{u}}^\ell[t])$;

14:    $\triangleright$ Calculate $\mathbf{s}^\ell[t]$ and $\mathbf{u}^\ell[t]$ according to Eq. (2~3);

15:    $\triangleright$ Perform the inference on the model $\mathcal{M}_{\text{tea}}$ based on Eq. (1~3) and record its membrane potential;

16: **end for**

17: $\mathcal{L}_{\text{MPFD}} = \sum_{\{\ell', \ell\}} \sum_t \left\| \mathbf{G}_{\text{tea}}^{\ell'}[t] - \mathbf{G}_{\text{stu}}^\ell[t] \right\|_2$;

18: Compute the loss: $\mathcal{L} = \mathcal{L}_{ce} + \lambda \mathcal{L}_{\text{MPFD}}$;

19: Backpropagation and update model parameters;

## 5 Experiment

In this section, we evaluate the performance of S$^2$NN on various tasks, including classification, object detection, and semantic segmentation. Then, we conduct ablation studies to verify the effect of the OS-Quant and MPFD. Details on experimental setups are provided in Appendix K.

### 5.1 Performance Comparison

**Image Classification**   We assess S$^2$NN on various architectures like MS-ResNet [49], VGGSNN [50], and spike-driven Transformer v3 (SDT3) [21], comparing it with advanced compression methods in SNNs, like BitSNN [16], Q-SNN [15], Q-Spikformer [51], and BESTformer [52]. Results in Tab. 1 reveal three conclusions. First, with $\eta = 6$ (W is 0.67 bit), S$^2$NN achieves SOTA results on all datasets, reducing size and OPs by $1.4\times \sim 6.8\times$. Second, despite significant reductions in size and OPs when $\eta = 4$, S$^2$NN outperforms the base on CIFAR-10 and ImageNet-1K, with only a small performance drop on other datasets. Third, S$^2$NN performs as well as FP SNN on simple datasets with fewer resources. Despite a gap on ImageNet-1K, it outperforms the advanced work [52] by 3.5%~4.6%. In Appendix F, we supplement our comparison with related BNN methods.

**Object Detection**   We use the COCO dataset to evaluate the efficacy of S$^2$NN in detection tasks. Following previous studies [53, 21], we use the mmdetection codebase, with S$^2$NN as the backbone for feature extraction and Mask R-CNN [54] for detection. The backbone is initialized using a network pre-trained on ImageNet-1K, with $\eta = 6$. Visualization and results are shown in Fig. 4(a)

Table 1: Performance of image classification. The colored values in brackets are the reduction factor of $S^2NN$ relative to the baseline.

| DATASET | METHOD | ARCITECTURE | BIT (W/A) | SIZE (MBIT) | OPS (G) | ACC. (%) |
|---|---|---|---|---|---|---|
| CIFAR-10 | FP SNN | RESNET-19 | 32/1 | 400.07 | 15.70 | 96.68 |
| | BITSNN [16] | RESNET-18 | 1/1 | 11.34 | - | 94.37 |
| | Q-SNN [15] | RESNET-19 | 1/1 | 13.04 BASE | 6.40 BASE | 95.54 BASE |
| | $S^2NN$ | RESNET-19 | 0.67/1 | 8.92 (1.5×) | 4.29 (1.5×) | 96.43 (+0.9) |
| | $S^2NN$ | RESNET-19 | 0.56/1 | 7.55 (1.7×) | 3.58 (1.8×) | 96.36 (+0.8) |
| | $S^2NN$ | RESNET-19 | 0.44/1 | 6.05 (2.2×) | 2.82 (2.3×) | 95.99 (+0.5) |
| CIFAR-100 | FP SNN | RESNET-19 | 32/1 | 401.55 | 18.89 | 80.42 |
| | Q-SNN [15] | RESNET-19 | 1/1 | 14.52 BASE | 6.50 BASE | 78.77 BASE |
| | $S^2NN$ | RESNET-19 | 0.67/1 | 10.40 (1.4×) | 4.39 (1.4×) | 78.77 (+0.0) |
| | $S^2NN$ | RESNET-19 | 0.56/1 | 9.03 (1.6×) | 3.67 (1.8×) | 78.43 (-0.3) |
| | $S^2NN$ | RESNET-19 | 0.44/1 | 7.53 (1.9×) | 2.88 (2.3×) | 77.40 (-1.4) |
| IMAGENET-1K | FP SNN | SDTV3-19M | 32/1 | 607.57 | 16.03 | 79.80 |
| | QSPIKF [51] | SPIKFORMER-8-512 | 1/1 | 36.80 | 2.12 | 54.54 |
| | BESTF [52] | SPIKFORMER-8-512 | 1/1 | 44.56 BASE | 5.67 BASE | 63.46 BASE |
| | $S^2NN$ | SDTV3-19M | 0.67/1 | 17.32 (2.6×) | 0.84 (6.8×) | 68.02 (+4.6) |
| | $S^2NN$ | SDTV3-19M | 0.56/1 | 15.88 (2.8×) | 0.78 (7.3×) | 67.43 (+4.0) |
| | $S^2NN$ | SDTV3-19M | 0.44/1 | 14.31 (3.1×) | 0.73 (7.8×) | 67.00 (+3.5) |
| DVSCIFAR-10 | FP SNN | VGGSNN | 32/1 | 296.60 | 1.97 | 82.3 |
| | Q-SNN [15] | VGGSNN | 1/1 | 10.91 BASE | 0.31 BASE | 81.6 BASE |
| | $S^2NN$ | VGGSNN | 0.67/1 | 7.86 (1.4×) | 0.20 (1.6×) | 82.0 (+0.4) |
| | $S^2NN$ | VGGSNN | 0.56/1 | 6.85 (1.6×) | 0.17 (1.8×) | 81.6 (+0.0) |
| | $S^2NN$ | VGGSNN | 0.44/1 | 5.74 (1.9×) | 0.13 (2.4×) | 81.3 (-0.3) |

and Tab. 2. We compare $S^2NN$ with advanced SNN and BNN models, and results reveal two conclusions. First, the $S^2NN$ achieves comparable results to FP SNNs while remarkably reducing resource cost. For instance, our mAP@0.5 is comparable to that of [21] (i.e., 41.8%), but saves 2.08× in model size and 3.27× in power consumption. Second, compared to BNN methods, $S^2NN$ exhibits SOTA results, surpassing the advanced work [55] by 7.4%.

**Semantic Segmentation** We use the ADE20K dataset to evaluate the efficacy of the $S^2NN$ in segmentation tasks. Following prior studies [53, 21], we use the mmsegmentation, with $S^2NN$ as the backbone for feature extraction and semantic FPN for segmentation. The initialization mirrors that of the detection task. Visualization and results are shown in Fig. 4(b) and Tab. 3. We compare $S^2NN$ with advanced SNNs and BNNs, and results reveal two conclusions. First, $S^2NN$ is far superior to methods in BNN, e.g., it surpasses the advanced work [56] by 17%~17.6%. Second, $S^2NN$ yields comparable performance to FP SNNs with fewer resources, sufficiently validating the efficacy of our model in segmentation tasks. For example, our method outperforms [53] by 1.1% in the MIoU metric and reduces the model size and power consumption by 34.6× and 19.8×, respectively.

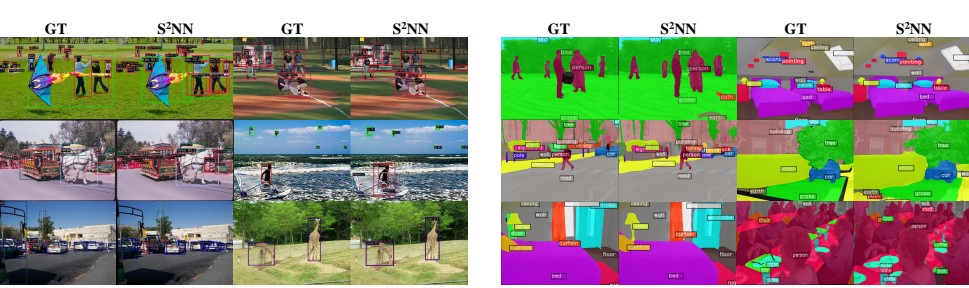

(a) Detection visualization of $S^2NN$.          (b) Segmentation visualization of $S^2NN$.

Figure 4: Detection and segmentation visualization of $S^2NN$ on COCO 2017 and ADE20K.

Table 2: Object detection results on COCO 2017.

| METHOD | BIT (W/A) | TIME STEP | SIZE (MBIT) | OPS (G) | MAP (@50%) |
|---|---|---|---|---|---|
| [57] | 32/1 | 64 | 547.2 | - | 33.1 |
| [58] | 32/1 | 7 | 803.2 | - | 41.9 |
| [53] | 32/1 | 1 | 2400 | - | 51.2 |
| | 32/1 | 1 | 1117 | - | 44.0 |
| [21] | 32/1 | 8 | 1238 | - | 58.8 |
| | 32/1 | 2 | 1238 | - | 41.8 |
| [59] | 1/1 | - | 10.99 | 0.55 | 31.0 |
| [60] | 1/1 | - | 175.0 | 3.22 | 32.9 |
| [61] | 1/1 | - | 173.3 | 3.21 | 37.2 |
| [55] | 1/1 | - | 15.52 | 0.6 | 32.6 |
| S$^2$NN | 0.67/1 | 8 | 595.5 | 0.84 | 40.0 |

Table 3: Segmentation results on ADE20K.

| METHOD | BIT (W/A) | TIME STEP | SIZE (MBIT) | POWER (MJ) | PIX ACC(%) | MIOU (%) |
|---|---|---|---|---|---|---|
| [29] | 1/1 | - | - | - | 62.8 | 9.22 |
| [62] | 1/1 | - | - | - | 59.5 | 7.16 |
| [63] | 1/1 | - | - | - | 59.5 | 9.74 |
| [56] | 1/1 | - | - | - | 67.3 | 18.8 |
| [53] | 32/1 | 1 | 528 | 22.1 | - | 32.3 |
| | 32/1 | 4 | 1914 | 183.6 | - | 35.3 |
| [21] | 32/1 | 4 | 352 | 27.2 | - | 40.1 |
| | 32/1 | 8 | 352 | 33.6 | - | 41.4 |
| S$^2$NN | 0.67/1 | 8 | 55.39 | 9.27 | 77.3 | 36.4 |
| S$^2$NN | 0.56/1 | 8 | 55.36 | 9.23 | 77.5 | 36.2 |
| S$^2$NN | 0.44/1 | 8 | 55.32 | 9.19 | 77.4 | 35.8 |

## 5.2 Ablation Study

We conduct ablation studies on the proposed OS-Quant and MPFD methods to demonstrate their effectiveness. Experiments are conducted on CIFAR-100 with $\eta = 6$ (weight set to 0.67 bit). The results are summarized in Tab. 4. First, we replace the sub-bit weight quantization in the baseline with the QS-Quant, resulting in a 0.48% performance improvement. This result underscores the effectiveness of OS-Quant. In addition, we compare three

Table 4: Ablation study of our S$^2$NN.

| BASELINE | OS-QUANT | LGKD | FRFD | MPFD | ACC. (%) |
|---|---|---|---|---|---|
| ✔ | - | - | - | - | 75.11 (BASE) |
| ✔ | ✔ | - | - | - | 75.59 (+0.48) |
| ✔ | ✔ | ✔ | - | - | 75.92 (+0.81) |
| ✔ | ✔ | - | ✔ | - | 76.71 (+1.60) |
| ✔ | ✔ | - | - | ✔ | 78.77 (+3.66) |

distillation methods mentioned in Sec. 4.3 to validate the effectiveness of MPFD. Specifically, the performance for LGKD, FRFD, and MPFD are 75.92%, 76.71%, and 78.77%, respectively. These results indicate that feature-based distillation outperform logit-based methods, and our MPFD achieves higher performance than firing rate-based feature distillation by providing more precise optimization directions. In conclusin, integrating the OS-Quant and MPFD into the baseline improves S$^2$NN's performance by 3.66%, underscoring their effectiveness.

## 6 Conclusion

SNNs have emerged as a promising paradigm for energy-efficient machine intelligence. However, as SNNs scale up to meet practical demands, their storage and computational requirements pose challenges for resource-constrained deployment. This work introduces S$^2$NN, a novel sub-bit compression framework for SNNs that represents weights with less than one bit. Through the introduction of OS-Quant and MPFD, S$^2$NN effectively addresses quantization bias and preserves performance. Experiments show that S$^2$NN achieves SOTA performance while significantly reducing model size and computational costs, making it particularly suitable for edge computing applications.

## Acknowledgments

This work is supported in part by the National Natural Science Foundation of China (No. 62220106008 and 62271432), in part by the Shenzhen Science and Technology Program (Shenzhen Key Laboratory, Grant No. ZDSYS20230626091302006) and in part by the Program for Guangdong Introducing Innovative and Entrepreneurial Teams, Grant No. 2023ZT10X044, in part by the State Key Laboratory of Brain Cognition and Brain-inspired Intelligence Technology, Grant No. SKLBI-K2025010. This work was partially supported by UESTC Kunpeng&Ascend Center of Cultivation.

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

# A    Ratio of Top-k codewords in Clustering Distribution

We analyze the top-k ratio in BSNN kernels for ResNet and VGG, with results shown Tab. 5. Both models show clustered distributions, with the clustering becoming more noticeable in deeper layers.

Table 5: The top-k ratio in BSNN kernels for ResNet and VGG structures.

| CIFAR-100, Res19 | Top2 | Top4 | Top8 | Top16 | Top32 | Top64 | Top128 |
|---|---|---|---|---|---|---|---|
| layer1.2 | 20.7% | 24.4% | 29.6% | 37.2% | 47.5% | 60.8% | 74.7% |
| layer2.2 | 21.8% | 24.8% | 30.0% | 38.6% | 50.2% | 65.0% | 80.0% |
| layer3.1 | 46.8% | 50.2% | 54.5% | 60.1% | 67.9% | 75.7% | 83.9% |
| DVSCIFAR10, VGGSNN | Top2 | Top4 | Top8 | Top16 | Top32 | Top64 | Top128 |
| conv3 | 15.1% | 18.1% | 23.6% | 31.8% | 43.2% | 59.1% | 74.7% |
| conv5 | 14.9% | 18.6% | 23.8% | 31.6% | 43.5% | 59.9% | 76.1% |
| conv7 | 14.2% | 18.0% | 24.6% | 34.1% | 47.7% | 65.8% | 82.3% |

# B    Analysis of Outlier Occurrence about Models, Datasets, and Augmentation

Fig 2(b) depicts a small example of the first 60 kernels from ResNet-19's second layer on CIFAR-100. To demonstrate that the emergence of outliers is a common phenomenon, we count the percentage of convolutional kernels containing outliers in each layer. Results are shown in Tab. 6, indicating that while the number of outlier-containing kernels varies by models, datasets, and augmentation, outlier occurrence remains a universal and severe issue.

Table 6: The percentage of convolutional kernels containing outliers.

| Configuration | layer1.1 | layer2.1 | layer3.1 |
|---|---|---|---|
| CIFAR10, Res19 | 21.15% | 17.62% | 22.33% |
| CIFAR-100, Res19 | 33.19% | 18.99% | 20.19% |
| CIFAR-100, Res19, only RandomCrop | 30.10% | 26.24% | 17.61% |
| CIFAR-100, Res19, only RandomHorizontalFlip | 29.16% | 21.61% | 29.90% |
| | conv3 | conv5 | conv7 |
| DVSCIFAR10, VGGSNN | 33.73% | 15.99% | 27.39% |

# C    Analysis of Hyperparameter $\gamma$ in OS-Quant

We test the performance of different $\gamma$ values on CIFAR-100 with ResNet-19 ($\eta$=6). Results are summarized as in Tab. 7, and the following conclusions are obtained:

- $\gamma$=0.5 (too low) treats too many values as outliers, causing excessive outlier scaling and destroying the kernel's spatial information.
- $\gamma$=3.0 (too high) fails to detect outliers effectively, degrading $S^2NN$ to baseline performance.
- $\gamma$=1.5 achieves optimal balance between outlier detection and spatial preservation.
- $\gamma$=1.0 and $\gamma$=2.0 perform well but slightly below $\gamma$=1.5.

Table 7: Accuracy under different $\gamma$ values.

| $\gamma$ | 0.5 | 1.0 | 1.5 | 2.0 | 3.0 |
|---|---|---|---|---|---|
| Accuracy | 74.48% | 75.34% | 75.59% | 75.19% | 75.08% |

# D    Comparison of OS-Quant with alternative outlier-handling methods

We test several outlier-handling methods against OS-Quant on CIFAR-100 using ResNet-19 ($\eta$=6):

- `Hamming distance`: Binarizes 32-bit kernels before calculating distance to codewords. This is the simplest method to solve codeword selection bias but fails to preserve spatial information.

- `Clip methods`: Instead of our IQR-based outlier detection and spatially-aware outlier scaling methods, we select three clipping variants. (1) $\texttt{Layer}_{\texttt{clip}}$: uses 1st/99th percentile of layer weights as clipping bounds; (2) $\texttt{IQR}_{\texttt{clip}}$: uses IQR for outlier detection and clipping; (3) $\texttt{Z-score}_{\texttt{clip}}$: uses Z-score for outlier detection and clipping.

- `Smooth operation`: Applied weight decay (1e-3 and 1e-4) during training.

- $\texttt{Z-score}_{\texttt{scale}}$: Combines Z-score outlier detection with our spatially-aware outlier scaling.

Table 8: Accuracy comparison of different outlier-handling methods.

|      | OS-Quant | Hamming | $\texttt{Layer}_{\texttt{clip}}$ | $\texttt{IQR}_{\texttt{clip}}$ | $\texttt{Z-score}_{\texttt{clip}}$ | 1e-3wd | 1e-4wd | $\texttt{Z-score}_{\texttt{scale}}$ |
|------|----------|---------|------------|----------|-------------|--------|--------|--------------|
| Acc. | 75.59    | 70.82   | 37.23      | 75.01    | 74.82       | 73.68  | 72.24  | 75.34        |

Experimental results are summarized in Tab. 8, which confirms the effectiveness of OS-Quant. Furthermore, we analyze the potential reasons for the poor performance of other methods.

- `Hamming distance`: Removes outliers via binarization but creates selection ambiguity. For example, given a full precision kernel $\mathbf{f}$ = [0.8, 0.7, 5, -0.9, -0.8, -0.7, -0.9, -0.8, -0.9], and two codewords $\mathbf{k}_1$ = [1, -1, 1, -1, -1, -1, -1, -1, -1] and $\mathbf{k}_2$ = [1, 1, 1, -1, -1, -1, 1, -1, -1]. $\mathbf{f}$ has equal Hamming distance, i.e., 1, to $\mathbf{k}_1$ and $\mathbf{k}_2$, causing selection ambiguity. In contrast, OS-Quant clearly differentiates distances (0.33 vs 3.93), identifying $\mathbf{k}_1$ as a better match. This shows the importance of preserving the spatial information of full-precision kernels when handling outliers.

- `Clip methods`: Fig 2(b) shows significant variation in kernel distributions, meaning the same value may be an outlier in some kernels but not others. Thus, using network-wide or layer-wide parameters to determine clipping thresholds for each kernel is inappropriate, as confirmed by $\texttt{Layer}_{\texttt{clip}}$. Other clipping methods perform better but still lag behind OS-Quant and $\texttt{Z-score}_{\texttt{scale}}$ due to their failure to preserve outliers' spatial information.

- `Smooth operation`: Based on our analysis of outlier distribution, this method can slightly mitigate outlier occurrence but slows convergence, thereby yielding lower performance with equal training epochs.

- $\texttt{Z-score}_{\texttt{scale}}$: Achieves top-2 results, but its effectiveness is limited because mean and variance calculations are influenced by the outliers themselves. Its performance gap with OS-Quant would widen on larger complex datasets.

# E    Detail Analysis of MPFD

In this section, we conduct a more detailed analysis of MPFD. Our main contribution to MPFD is performing distillation at the membrane potential level, providing more precise gradient guidance for highly compressed models. That is, directly using the 2-norm to calculate membrane potential errors between teachers and students can also provide more accurate gradient guidance compared to traditional FRFD and LGKD. Therefore, if your goal is to avoid introducing excessive computation during training, you can choose to use simpler error calculation methods rather than the Gram matrix. In the following, we will mainly discuss the impact of the Gram matrix in the MPFD method on the performance, as well as the advantages and disadvantages of using and not using the Gram matrix.

We first evaluate membrane potential-based distillation performance with and without the Gram matrix and other approaches. LGKD makes the student mimic the teacher network's final logits (the raw output values before the softmax activation function is applied) to transfer knowledge [43]. FRFD is a classic feature distillation scheme in the SNN domain [47], which uses the firing rate as

the intermediate features of the network and align the firing rates between teacher and student models. Therefore, neither LGKD nor FRFD uses a Gram matrix. Experiments are conducted on CIFAR-100 with ResNet-19. As shown in Tab. 9, the 2-norm membrane potential distillation achieves 78.32% accuracy, which demonstrates that direct membrane potential distillation can also offer more precise gradient guidance than FRFD, thus improving accuracy.

Table 9: Performance comparison of different knowledge distillation methods in sub-bit SNN.

|  | LGKD | FRKD | MPFD(w/o Gram) | MPFD(w/ Gram) |
|---|---|---|---|---|
| Accuracy | 75.92% | 76.71% | 78.32% | 78.77% |

Both `MPFD (w/o Gram)` and `MPFD (w/ Gram)` perform distillation on membrane potentials, offering more precise gradient guidance. Their respective pros and cons are as follows:

- `MPFD (w/ Gram)`. Gram matrix is typically regarded as capturing semantic relationships, so MPFD is usually correctly classified with higher confidence, leading to superior performance. This facilitates cross-architecture distillation without requiring matched network layers or identical dimensions. However, it incurs small additional computational costs.

- `MPFD (w/o Gram)`. It offers simpler implementation and lower computational costs. However, it has slightly lower performance than `MPFD (w/ Gram)`, and it doesn't capture the semantic relationships between features that the Gram matrix version does.

## F    Supplementary Comparison with BNNs on Image Classification task

We supplement the comparison with related methods in the BNN domain on the image classification task. The experimental results are summarized in the Table 10. These results demonstrate that $S^2NN$ performs competitively against existing BNN methods on static datasets. Specifically, when compared to the sub-bit neural network [23] that also operates with weights below 1-bit, $S^2NN$ achieves notable accuracy improvements of 5%-6% on CIFAR-10 and 6%-9% on ImageNet-1K. Notably, $S^2NN$ outperforms the sub-bit neural network, even when the latter employs 32-bit activations. Furthermore, when compared to conventional BNNs with 1-bit weights, $S^2NN$ shows superior performance on CIFAR-10 using sub-1-bit weights, while maintaining competitive accuracy with state-of-the-art methods on ImageNet-1K. These comprehensive results, along with those presented in Table 1, decisively validate the effectiveness of $S^2NN$.

## G    Performance Improvement Analysis about OS-Quant and MPFD

OS-Quant improves performance in three ways:

- **Observing the codeword selection bias**, which is the quantization error that typically causes performance degradation. Addressing it will yield accuracy improvements.

- **Using IQR for outliers detection**, which remains reliable even with limited data and isn't influenced by extreme values.

- **Implementing spatially-aware scaling**, which effectively eliminates outlier interference while preserving crucial kernel spatial features for proper codeword selection.

MPFD improves performance by applying distillation at the membrane potential level, enabling more precise optimization than firing rate-based distillation.

## H    Model Compression & Acceleration

**Compression**    We explain in detail how $S^2NN$ achieves sub-1-bit model compression. For simplicity, we analyze the parameter storage of a single layer in the model. Consider a layer with $c_{out} \times c_{in} \times k_w \times k_h$ parameters. As shown in the 'kernel storage' on the left side of Figure 5, standard BSNN requires 1 bit to store each weight parameter, resulting in a total storage requirement of $c_{out} \times c_{in} \times k_w \times k_h$

Table 10: Supplementary comparison with related BNN methods on the image classification task.

| Dataset | Method | Arcitecture | Sub-bit | Weight Bit | Activity Bit | Accuracy (%) |
|---|---|---|---|---|---|---|
| CIFAR-10 | IR-Net [28] [CVPR20] | Res18 | ✗ | 1 | 32 | 92.9 |
| | | Res18 | ✔ | 0.67 | 32 | 92.7 |
| | SNN [23] [ICCV2021] | Res18 | ✔ | 0.56 | 32 | 92.3 |
| | | Res18 | ✔ | 0.44 | 32 | 91.9 |
| | IR-Net [28] [CVPR20] | Res18 | ✗ | 1 | 1 | 91.5 |
| | | Res18 | ✔ | 0.67 | 1 | 91.0 |
| | SNN [23] [ICCV21] | Res18 | ✔ | 0.56 | 1 | 90.6 |
| | | Res18 | ✔ | 0.44 | 1 | 90.1 |
| | ProxConnect++ [64] [NeurIPS23] | Res20 | ✔ | 1 | 1 | 90.2 |
| | A&B[65] [CVPR24] | ReAct18 | ✗ | 1 | 1 | 92.3 |
| | S$^2$NN | Res19 | ✔ | 0.67 | 1 | 96.43 |
| | S$^2$NN | Res19 | ✔ | 0.56 | 1 | 96.36 |
| | S$^2$NN | Res19 | ✔ | 0.44 | 1 | 95.99 |
| ImageNet-1K | IR-Net [28] [CVPR20] | Res34 | ✗ | 1 | 32 | 70.4 |
| | | Res34 | ✔ | 0.67 | 32 | 68.0 |
| | SNN [23] [ICCV21] | Res34 | ✔ | 0.56 | 32 | 66.9 |
| | | Res34 | ✔ | 0.44 | 32 | 65.1 |
| | Bi-Real [27] [IJCV20] | Res34 | ✗ | 1 | 1 | 62.2 |
| | IR-Net [28] [CVPR20] | Res34 | ✗ | 1 | 1 | 62.9 |
| | | Res34 | ✔ | 0.67 | 1 | 61.4 |
| | SNN [23] [ICCV21] | Res34 | ✔ | 0.56 | 1 | 60.2 |
| | | Res34 | ✔ | 0.44 | 1 | 58.6 |
| | BiBert [66] [ICLR22] | Swin-T | ✗ | 1 | 1 | 68.3 |
| | BinaryViT [67] [CVPR23W] | ViT | ✗ | 1 | 1 | 67.7 |
| | ProxConnect++ [64] [NeurIPS23] | ViT-B | ✗ | 1 | 1 | 66.3 |
| | A&B[65] [CVPR24] | ReActA | ✗ | 1 | 1 | 66.9 |
| | S$^2$NN | SDT3 | ✔ | 0.67 | 1 | 68.02 |
| | S$^2$NN | SDT3 | ✔ | 0.56 | 1 | 67.43 |
| | S$^2$NN | SDT3 | ✔ | 0.44 | 1 | 67.00 |

bits per layer. In contrast, S$^2$NN requires fewer than $c_{out} \times c_{in} \times k_w \times k_h$ bits to achieve more efficient storage. Specifically, as described in Section 3, S$^2$NN performs forward propagation using a compact codebook $\mathbb{P}$ rather than a full codebook. This allows S$^2$NN to achieve compression below 1-bit by storing two components: (1) *the indices of each kernel parameter in $\mathbb{P}$*, and (2) *the mapping relationship between indices and weights*. For the first component, S$^2$NN needs to store the indices of each kernel parameter in the compact codebook, with a total number of $c_{out} \times c_{in}$ kernels (also indices). Since the compact codebook contains $2^\eta$ binary codewords, the indices range from 1 to $2^\eta$, requiring $\eta$ bits per index. Therefore, representing the indices for this layer's parameters requires $c_{in} \times c_{out} \times \eta$ bits. For the second component, S$^2$NN involves storing the compact codebook $\mathbb{P}$, which contains $2^\eta$ elements, and each element is a $k_w \times k_h$ binary kernel. Thus, storing $\mathbb{P}$ requires $2^\eta \times k_w \times k_h$ bits. Therefore, the total storage requirement for S$^2$NN is $c_i \times c_o \times \eta + 2^\eta \times k_w \times k_h$ bits. Compared to BSNN, S$^2$NN achieves a compression ratio of $\frac{c_{out} \times c_{in} \times \eta + 2^\eta \times k_w \times k_h}{c_{out} \times c_{in} \times k_w \times k_h}$. Given that $\eta < k_w \times k_h$, this ratio approximates to $\frac{\eta}{k_w \times k_h}$.

**Acceleration** In addition to model compression, we also discuss the hardware-friendly characteristics of S$^2$NN. Benefiting from the advantages of model compression above, the hardware implementation of S$^2$NN is theoretically more efficient than that of the standard BSNN. Specifically,

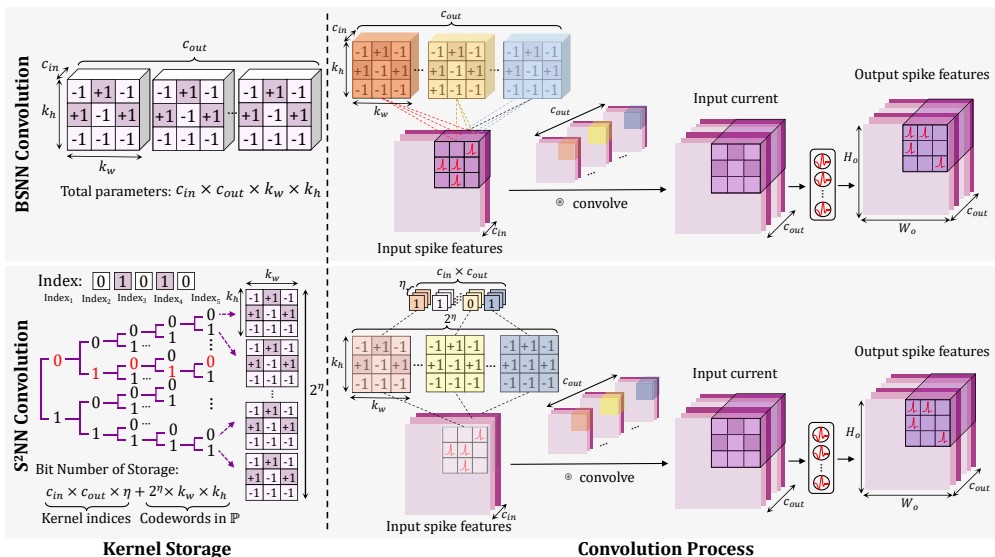

Figure 5: Comparison of compression and acceleration between standard BSNN and our $S^2$NN during the convolution process.

the weight transmission volume in each layer of $S^2$NN are significantly lower than those of BSNN, resulting in a substantial reduction in data movement between on-chip and off-chip. This reduction yields three key improvements in hardware efficiency: first, it significantly lowers data transmission latency between on-chip and off-chip, thereby accelerating the inference process; second, it reduces the demand for on-chip storage, minimizing memory overhead. When $\eta$ is set to 4, the on-chip storage requirement can be saved by approximately $\frac{5}{9}$ compared to the standard BSNN; third, it decreases the number of off-chip memory accesses, which can lower energy consumption. Between on-chip and off-chip data movement is typically one of the most power-hungry operations [68]. By maximizing the reuse of weights in $\mathbb{P}$, the movement of data between on-chip and off-chip is minimized, leading to a more energy-efficient design. The outstanding features demonstrated by $S^2$NN may inspire and drive algorithm-driven chip design, while simultaneously reducing the algorithm exploration costs required prior to hardware design.

# I  Hardware Validation

We compare $S^2$NN and BSNN on an FPGA to quantify $S^2$NN's advantages. Experimental Settings:

- Platform: Xilinx Vivado 2021.2

- Simulation Platform: Modelsim

- Clock Frequency: 100MHz

- AXI Bit Width: 32 bits

- Model: SCNN: 32×32-64c3-128c3-128c3-256c3-256c3-256c3-10, T=8, $\eta$=5

We measure $S^2$NN and BSNN's per-layer data access and latency between on-chip and off-chip memory. As shown in Tab. 11, despite additional indices, $S^2$NN achieves lower data access due to sub-bit compression (columns 2-3), and also reduces on/off-chip transmission latency (columns 4-5). **Compared to FPSNN, $S^2$NN's advantages are even more obvious.** We also present the Modelsim simulation results for the 5th layer in Fig 6, showing data access and access latency for $S^2$NN and BSNN. Notably, the benefits of sub-bit weight transmission are significant relative to the codebook overhead. As per [69], off-chip DRAM access requires 128× more energy than on-chip SRAM access and 6400× more than integer addition. Compared to BSNN, $S^2$NN reduces DRAM access by 3.6× (Columns 2-3), thereby leading to highly significant energy savings benefits.

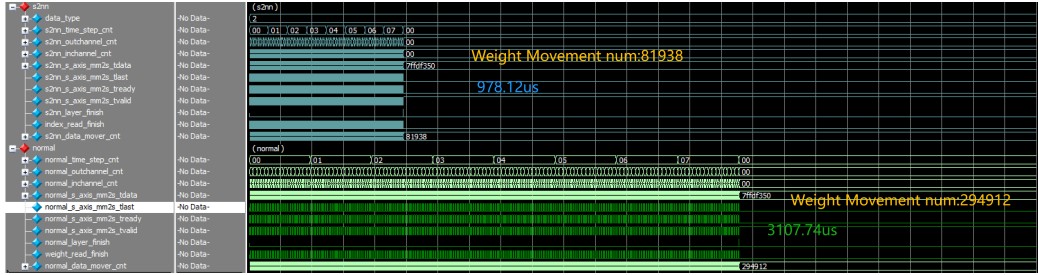

Figure 6: Modelsim simulation results for the 5th layer, including data access and access latency for S$^2$NN (top) and BSNN (bottom).

Table 11: Hardware validation of S$^2$NN and BSNN.

| | Data Access | | Access Latency ($\mu s$) | |
|---|---|---|---|---|
| | S$^2$NN | BSNN | S$^2$NN | BSNN |
| Layer1 | 5138 | 18432 | 61.32 | 194.14 |
| Layer2 | 10258 | 36864 | 122.44 | 388.38 |
| Layer3 | 20498 | 73728 | 244.68 | 776.86 |
| Layer4 | 40978 | 147456 | 489.17 | 1553.82 |
| Layer5 | 81938 | 294912 | 978.12 | 3107.74 |
| Layer6 | 81938 | 294912 | 978.12 | 3107.74 |

# J    Energy Consumption Calculation

We use the standard SNN energy calculation [70]: $E_{total} = E_{MAC} \cdot FLOPs^1_{conv} + E_{AC} \cdot (\sum_{n=2}^{N} SOPs^N + \sum_{m=1}^{M} SOPs^M)$, where both $SOPs = fr \cdot T \cdot FLOPs$ and $E_{AC}$ are bit-width dependent. Most SNN research uses 45nm technology for energy calculations. After investigation, binary weights reduce SOPs to 1/64 of fp32 values[71], but the literature lacks $E_{AC}$ for binary operations at 45nm. For fair comparison with existing work, we use fp32-based $E_{AC}$ at 45nm (0.9pJ). As a result, our actual energy is lower than reported.

# K    Experiment Details

## K.1    Image Classification

**Dataset**    CIFAR-10 [72] is a widely used computer vision dataset that contains 10 categories, with 6,000 32×32 pixel color images per category, totaling 60,000 images. CIFAR-100 [72] maintains identical image dimensions and total count, containing 100 fine-grained categories grouped into 20 superclasses. ImageNet-1K [73] is a large-scale visual database comprising over 1.2 million training images and 50,000 validation images across 1,000 object categories. Its extensive category coverage and image diversity have established ImageNet-1K as a pivotal benchmark dataset in deep learning and computer vision research. DVS-CIFAR10 [74] consists of 10,000 event streams generated by converting the original CIFAR-10 images using an event-based sensor with a resolution of 128×128 pixels. The dataset preserves the original 10 categorization structure. These datasets hold substantial importance within machine learning and neuromorphic computing, serving as standard benchmarks for evaluating diverse methodologies.

**Setup**    We conduct three experiments across all datasets. For convolutional layers with kernel size greater than 1, we set the cardinality of the compact codebook $\mathbb{P}$ to 16, 32, and 64, corresponding to parameter $\eta$ values of 4, 5, and 6, respectively. Noteworthy, our S$^2$NN method can also be extended to convolutions with a kernel size of 1. Since 1×1 convolutions already have relatively low computational complexity and parameter count, we do not apply further compression to these kernels. In our experiments, we employ the Spike-driven Transformer v3 model with 19M parameters, which uses a time step of 1 during training and 4 during inference. For the CIFAR-10 and CIFAR-100

datasets, we adopt MS-ResNet with a time step of 6, following prior work [75, 49]. In contrast, Q-SNN and BitSNN use SpikingResNet, which typically uses fewer time steps, such as 2 or 4. We employ MS-ResNet due to its advanced membrane potential-based residual connections. Additionally, we supplement experiments with SpikingResNet, achieving 95.56% accuracy on CIFAR-10 and 78.51% on CIFAR-100 with a bit-width of 0.44 and time step 2. In our experiments, we employ the full-precision counterpart as the teacher model. The detailed hyperparameter settings are provided in Table 12.

Table 12: Hyper-parameters for image classification.

| Hyper-parameter | ImageNet-1K | CIFAR-10 | CIFAR-100 | DVS-CIFAR10 |
|---|---|---|---|---|
| Timestep | 1×4 | 6 | 6 | 10 |
| Epochs | 200 | 250 | 250 | 300 |
| Resolution | 224×224 | 32×32 | 32×32 | 48×48 |
| Batch size | 1024 | 128 | 128 | 32 |
| Optimizer | Adam | Adam | Adam | Adam |
| Weight decay | 0 | 0 | 0 | 0 |
| Initial learning rate | 6e-4 | 5e-4 | 5e-4 | 5e-4 |
| Learning rate decay | Cosine | Cosine | Cosine | Cosine |
| Warmup epochs | 10 | None | None | None |
| Label smoothing | 0.1 | None | None | None |

## K.2 Object Detection

**Dataset** COCO 2017 [76] is a large-scale computer vision dataset designed for multiple tasks, including object detection, segmentation, and image captioning. The dataset consists of 118,287 training images, 5,000 validation images, and 40,670 test images. It provides multiple types of annotations, including object detection annotations (covering 80 common object categories), instance segmentation masks (detailing the contours of each object), and natural language annotations with five descriptive sentences per image. COCO emphasizes contextual relationships between objects in everyday scenes, making it a crucial benchmark for evaluating computer vision algorithms in practical applications.

**Setup** In the COCO experiment, similar to previous work [53, 21], we first convert the *mmdetection* codebase to the spike version and then use it for our experiments. We employ our highly compressed $S^2NN$ as the backbone and use Mask R-CNN as the detector to obtain the final model. The backbone is initialized with the weights of the $S^2NN$ ($\eta = 6$) pre-trained on ImageNet-1K, while the additional layers are initialized using Xavier [77] initialization. We fine-tune the model for 30 epochs on the COCO dataset. During fine-tuning, we resize and crop both the training and test images to $1333 \times 800$. Additionally, we apply random horizontal flipping and resize the training images with a ratio of 0.5. The batch size is set to 16. We use the AdamW optimizer with an initial learning rate of 1e-4, and the learning rate decays according to a polynomial schedule with an exponent of 0.9. The results of our method on the COCO dataset are shown in Figure 4(a).

## K.3 Semantic Segmentation

**Dataset** ADE20K is a comprehensive semantic segmentation dataset widely used in computer vision research. It contains over 20,000 images spanning a diverse range of indoor and outdoor scenes, with pixel-level annotations for 150 object and stuff categories, such as person, tree, and sky. ADE20K covers a variety of challenging environments, including urban areas, streets, buildings, and natural landscapes, making it ideal for training models to recognize and segment complex scenes. The dataset is particularly valuable for evaluating semantic segmentation algorithms, as it provides detailed ground truth annotations, including both object categories and background elements.

**Setup** Following our object detection experiments, we convert the *mmsegmentation* [78] codebase to its spike version and use it for our experiments. Similar to the object detection task experiments, we

employ our highly compressed $S^2NN$ as the backbone for feature extraction, integrated with Semantic FPN [79] for segmentation. In this task, we conduct three experiments using $S^2NN$ pre-trained on ImageNet-1K with $\eta$ values of 4, 5, and 6 for the backbone initialization. The newly added layers are initialized using Xavier initialization [77]. During training, we use the AdamW optimizer with an initial learning rate of $1 \times 10^{-4}$ that follows a polynomial decay schedule with an exponent of 0.9. We train for 160K iterations with a batch size of 16, incorporating a linear warm-up period during the first 1500 iterations. The results of our method on the ADE20K dataset are shown in Figure 4(b).

## L  Novelty of $S^2NN$

$S^2NN$ is an incremental innovation based on existing research, with contributions in three key aspects.

- **SNN domain**. First, we pioneer introducing the sub-bit concept and realize below 1-bit SNN models. This provides a potential solution for the deployment of SNN at edge devices and the future development of neuromorphic hardware. Second, we provide a MPFD, which offers more accurate gradient guidance than existing feature distillation, improving highly compressed SNN performance.

- **Model compression domain**. We first identify the 'Outlier-induced Codeword Selection Bias' and propose OS-Quant to address the quantization error caused by outliers, improving accuracy and convergence.

- **Comprehensive evaluation**. We extensively evaluate $S^2NN$ on diverse architectures, vision, and NLP tasks, establishing a new comparative benchmark lacking in previous SNN compression research.

