# OpenReview forum: "S$^2$NN: Sub-bit Spiking Neural Networks"
_NeurIPS.cc/2025/Conference — NeurIPS 2025 poster_

### Official Review · Reviewer_LxTv · 2025-06-27

**Clarity:** 3
**Significance:** 4
**Originality:** 4
**Rating:** 5
**Confidence:** 5

**Summary:**

The paper introduces Sub-bit Spiking Neural Networks (S²NNs), a framework for compressing and accelerating SNNs by representing weights with less than one bit. Extensive experiments on vision and non-vision tasks demonstrate that S²NN achieves state-of-the-art performance and efficiency, making it suitable for edge computing applications.

**Questions:**

1.The OS-Quant method detects and scales outliers based on spatial neighbors. How does this approach perform in cases where outliers are spatially isolated or when kernel sizes vary significantly (e.g., 1x1 vs. 3x3 or others)?


2.The paper compares S²NN to BSNNs and FP-SNNs. How does it fare against state-of-the-art quantized ANNs in terms of performance and efficiency, especially for similar bit-widths?

3.The results on NLP tasks (e.g., GLUE benchmark) show a performance drop compared to vision tasks. What are the key challenges in adapting S²NN to non-vision domains?

**Ethical Concerns:**

["NO or VERY MINOR ethics concerns only"]

**Final Justification:**

Thanks for response.

**Limitations:**

Yes

**Paper Formatting Concerns:**

No.

**Quality:**

3

**Strengths And Weaknesses:**

Strengths:

1.The paper introduces the novel concept of sub-bit weight representation in SNNs, pushing the boundaries of model compression and efficiency. The proposed OS-Quant and MPFD methods are well-motivated and address critical challenges in SNN quantization and performance preservation.

2.The paper validates S²NN across diverse tasks (classification, object detection, segmentation, and NLP) and architectures, demonstrating its scalability and robustness. This paper also performs additional hardware verification.

Weaknesses:

1.The reliance on clustering patterns in binary kernels may not hold universally across all architectures or tasks.

2.The MPFD method, while effective, introduces additional computational overhead during training, which could be a limitation for resource-constrained scenarios.

3.The comparison with other sub-bit quantization methods (e.g., in non-spiking networks) is limited.

---

> ### Author Rebuttal · Authors · 2025-07-31
>
> Dear reviewer LxTv, thank you for your review. Your concerns regarding our work primarily focus on **five** aspects: **(1) kernel clustering universality, (2) additional energy consumption for distillation, (3) how does OS-Quant perform in cases in informal scene, (4) the comparison with methods in ANNs, and (5) key challenges in adapting S²NN to non-vision domains.** We have provided detailed explanations for each of these concerns and hope to help resolve them.
>
> # W1: Limitation of kernel clustering universality.
>
> Yes, you raise a valid point. Indeed, the clustering patterns in binary kernels are more pronounced in vision architectures (i.e., image data) due to the prior of spatial correlation in convolutional filters. This phenomenon is less evident in sequential or text data, where structural assumptions differ.
> But luckily, our sub-bit compression method also delivers satisfactory performance when applied to non-image data. To demonstrate the effectiveness of our method in non-image tasks, we conduct additional experiments in the NLP task.
>
> *Experimental details*: We extend S²NN to SpikeLM[1] using the method in Appendix A.7 (η=5). We select a BERT architecture featuring a 12-layer encoder transformer within SpikeLM and conduct experiments on the GLUE benchmark. All other training parameters are maintained according to the original paper.
>
> *Experimental results*: The results are summarized in the table below. Clearly, our method maintains the satisfactory performance of 73.7%. **These results fully validate the effectiveness of S²NN for complex non-image tasks.**
>
> |         | SST-2 | MRPC | RTE  | MNLI$_m$ | QNLI | QQP$_F1$ | CoLA | STS-B | Avg. |
> | ------- | ----- | ---- | ---- | -------- | ---- | -------- | ---- | ----- | ---- |
> | SpikeLM | 87.0  | 85.7 | 69.0 | 77.1     | 85.3 | 83.9     | 38.8 | 84.9  | 76.5 |
> | S²NN    | 85.1  | 84.4 | 61.8 | 74.3     | 83.6 | 84.0     | 31.8 | 84.4  | 73.7 |
>
> *[1] SpikeLM: Towards General Spike-Driven Language Modeling via Elastic Bi-Spiking Mechanisms. ICML2024.*
>
> # W2. Additional energy consumption for distillation in edge scenarios.
>
> MPFD introduces additional computational overhead during training, but **zero overhead during deployment.** S²NN is designed to optimize SNN deployment efficiency while maintaining high performance. By quantizing the weight to sub-bit, S²NN achieves extreme model compression. Without distillation, S²NN will suffer from severe performance degradation and even convergence failures in challenging tasks. Thus, distillation is essential for preserving S²NN's accuracy. **Notably, distillation operates exclusively during training and remains fully decoupled from inference. This ensures no additional inference overhead, aligning with our design goal for S²NN.**
>
> # Q1. How does OS-Quant perform in cases where outliers are spatially isolated or when kernel sizes vary significantly?
>
> We appreciate this insightful question regarding outlier handling. The concern about spatially isolated outliers can be addressed through two key considerations.
>
> - First, for conventional kernels (kernel size >1), the inherent spatial connectivity of the receptive field ensures that all weights have adjacent neighbors, making spatially isolated outliers impossible by definition.
> - Second, for 1×1 kernels, we use the method in Appendix A.7 to aggregate adjacent 1×1 kernels into a large kernel, and then perform sub-bit compression, thereby creating meaningful neighbor relationships for all values, including potential outliers.
>
> In addition, the OS-Quant quantization method is designed to be architecture-agnostic, capable of handling any kernel, regardless of the size and change of the kernel size.
>
>
> # Q2 & W3. The comparison with related methods in the ANN domain is limited, especially for similar bit-widths.
>
> We supplement the comparison with related methods in the BNN domain on the image classification task. The experimental results are summarized in the table below. These results demonstrate that S²NN performs competitively against existing BNN methods on static datasets. Specifically, **when compared to the sub-bit neural network that also operates with weights below 1-bit, S²NN achieves notable accuracy improvements of 5%-6% on CIFAR-10 and 6%-9% on ImageNet-1K.** Notably, S²NN outperforms the sub-bit neural network, even when the latter employs 32-bit activations. Furthermore, **when compared to conventional BNNs with 1-bit weights, S²NN shows superior performance on CIFAR-10 using sub-1-bit weights while maintaining competitive accuracy with state-of-the-art methods on ImageNet-1K.** These comprehensive results, along with those presented in Table1, decisively validate the effectiveness of S²NN.
>
> ### CIFAR-10
>
> | Method                       | Arc.    | Weight Bit | Activity Bit | Acc. (%)  |
> | ---------------------------- | ------- | ---------- | ------------ | --------- |
> | IR-Net (CVPR 2020)           | Res18   | 1          | 32           | 92.9      |
> | SNN (ICCV 2021)              | Res18   | 0.67       | 32           | 92.7      |
> | SNN (ICCV 2021)              | Res18   | 0.56       | 32           | 92.3      |
> | SNN (ICCV 2021)              | Res18   | 0.44       | 32           | 91.9      |
> | IR-Net (CVPR 2020)           | Res18   | 1          | 1            | 91.5      |
> | SNN (ICCV 2021)              | Res18   | 0.67       | 1            | 91.0      |
> | SNN (ICCV 2021)              | Res18   | 0.56       | 1            | 90.6      |
> | SNN (ICCV 2021)              | Res18   | 0.44       | 1            | 90.1      |
> | ProxConnect++ (NeurIPS 2023) | Res20   | 1          | 1            | 90.2      |
> | A&B (CVPR 2024)              | ReAct18 | 1          | 1            | 92.3      |
> | **S²NN**                     | Res19   | 0.67       | 1            | **96.43** |
> | **S²NN**                     | Res19   | 0.56       | 1            | **96.36** |
> | **S²NN**                     | Res19   | 0.44       | 1            | **95.99** |
>
> ### ImageNet-1K
>
> | Method                       | Arc.   | Weight Bit | Activity Bit | Acc. (%)  |
> | ---------------------------- | ------ | ---------- | ------------ | --------- |
> | IR-Net (CVPR 2020)           | Res34  | 1          | 32           | 70.4      |
> | SNN (ICCV 2021)              | Res34  | 0.67       | 32           | 68.0      |
> | SNN (ICCV 2021)              | Res34  | 0.56       | 32           | 66.9      |
> | SNN (ICCV 2021)              | Res34  | 0.44       | 32           | 65.1      |
> | Bi-Real (IJCV 2020)          | Res34  | 1          | 1            | 62.2      |
> | IR-Net (CVPR 2020)           | Res34  | 1          | 1            | 62.9      |
> | SNN (ICCV 2021)              | Res34  | 0.67       | 1            | 61.4      |
> | SNN (ICCV 2021)              | Res34  | 0.56       | 1            | 60.2      |
> | SNN (ICCV 2021)              | Res34  | 0.44       | 1            | 58.6      |
> | BiBert (ICLR 2022)           | Swin-T | 1          | 1            | 68.3      |
> | Binary ViT (CVPR 2023W)      | ViT    | 1          | 1            | 67.7      |
> | ProxConnect++ (NeurIPS 2023) | ViT-B  | 1          | 1            | 66.3      |
> | A&B (CVPR 2024)              | ReActA | 1          | 1            | 66.9      |
> | **S²NN**                     | SDT3   | 0.67       | 1            | **68.02** |
> | **S²NN**                     | SDT3   | 0.56       | 1            | **67.43** |
> | **S²NN**                     | SDT3   | 0.44       | 1            | **67.00** |
>
> # Q3. What are the key challenges in adapting S²NN to non-vision domains?
>
> The key challenge in applying S²NN to non-vision domains primarily lies in the **weaker kernel clustering patterns in these architectures compared to vision models**. Due to local receptive fields and translation invariance, vision models naturally exhibit strong spatial clustering in their weight distributions. However, non-vision models lack such an inherent prior assumption, resulting in less pronounced clustering patterns. Nevertheless, our experiments on NLP tasks (Response to W1/Appendix A.8) demonstrate that even with these weaker clustering patterns, S²NN can still achieve an accuracy of 73.7% through its outlier-aware sub-bit quantization and membrane potential-based distillation techniques. This suggests that S²NN retains some cross-domain adaptability despite this architectural limitation.

---

> > ### Comment · Reviewer_LxTv · 2025-08-04
> > **The clarifications provided have addressed all my earlier concerns.**
> >
> > Thanks for your prompt reply. The clarifications provided have addressed all my earlier concerns, and I now have a clearer understanding of the work. Appreciate your efforts in elaborating on those points.

---

> ### Author Response · Authors · 2025-08-06
>
> Dear Reviewer LxTv,
>
> We are glad that your concerns have been addressed. We sincerely appreciate your recognition of our work. We will revise the manuscript carefully based on your suggestions, as well as those from the other reviewers, to further improve its quality.

---

### Official Review · Reviewer_WU5Y · 2025-06-30

**Clarity:** 4
**Significance:** 3
**Originality:** 3
**Rating:** 5
**Confidence:** 5

**Summary:**

This paper proposes a sub-bit Spiking Neural Network framework that compresses weights to below 1-bit. The authors build upon clustering patterns in binary kernels and introduce an OS-Quant method to reduce the codeword selection bias caused by outliers. Additionally, a MPFD approach is proposed to guide the highly compressed model more effectively. The paper provides extensive experimental evidence across classification, detection, and segmentation tasks and includes FPGA hardware validations, demonstrating significant improvements in both performance and efficiency compared to state-of-the-art methods.

**Questions:**

1. Could you provide more details about how the compact codebook is initialized? Is it random or based on some clustering of pretrained weights?

2. Is the codebook generation method sensitive to initialization or data distribution across datasets?

3. For practical deployment, what would be a good η value that balances accuracy and efficiency? Does this vary much across different tasks? Could you provide more details about how it is determined for different layers?

4. Could you elaborate on how the outlier detection threshold (γ=1.5) is chosen? Are there any sensitivity analyses performed?

5. Are the outliers detected by OS-Quant common across different tasks and structures? Can they be determined in advance?

**Ethical Concerns:**

["NO or VERY MINOR ethics concerns only"]

**Final Justification:**

My questions have been resolved. Since my initial rating was positive, I will maintain my rating and recommend acceptance.

**Limitations:**

Yes

**Paper Formatting Concerns:**

The article has no Paper Formatting Concerns.

**Quality:**

3

**Strengths And Weaknesses:**

Strengths:

1. For the SNN domain, this paper proposes a novel sub-bit SNN compression approach that significantly reduces model size and computational cost.

2. Includes hardware-oriented results, confirming the efficiency and deployment benefits of S2NN on FPGA platforms.

Weaknesses:

1. The paper does not provide sufficient analysis of model behavior at extremely low bit-width settings (η < 4), leaving questions about the practical lower limits of the proposed compression approach.

2. The paper lacks formal theoretical analysis explaining why sub-bit quantization is particularly effective, missing an opportunity to provide deeper insights into the underlying mechanisms.

3. The paper lacks detailed description of the important component compact codebookd, such as how it is initialized.

---

> ### Author Rebuttal · Authors · 2025-07-31
>
> Dear reviewer WU5Y, thank you for your review. Your concerns regarding our work primarily focus on **seven** aspects: **(1) effectiveness of S²NN at η < 4, (2) why OS-Quant is effective, (3) initialization of compact codebook, (4) is the codebook generation sensitive to initialization or data distribution across datasets, (5) the suitable value of η, (6) analysis of the outlier detection threshold γ, and (7) generalizability of outlier occurrence.** We have provided detailed explanations for each of these concerns and hope to help resolve them.
>
> # W1: Effectiveness of S²NN at extremely low bit-widths (η < 4).
>
> When η=4, the convolution kernel (e.g., 3*3 kernel) can only take 2⁴ different values. **This configuration reduces the codebook size from 2⁹ (full codebook) to 2⁴ (compact codebook), achieving a reduction of approximately 97%. Therefore, the configuration with η = 4 can be considered an extremely low-bit width setting**. To explore S²NN's potential at η<4 regimes, we conduct a supplementary experiment on CIFAR100 using ResNet-19 with η=3 (8 codewords, 0.33 bits/weight). As shown below, with only 8 codewords, S²NN demonstrates a remarkable 98.44% compression rate in kernel space representation. While this ultra-compact configuration achieves significant efficiency, we observe there's a noticeable performance impact (from 77.4% to 70.31%) due to the severely limited convolution kernel representation. Thus, we recommend maintaining η above 4 to ensure adequate representational capacity.
>
> | η        | Kernel Space        | Compression level of kernel space | Bit per weight | Accuracy (CIFAR100, ResNet19) |
> | -------- | ------------------- | --------------------------------- | -------------- | ----------------------------- |
> | 9 (BSNN) | 512 (Full codebook) | 0%                                | 1.00           | 78.77%                        |
> | 6        | 64                  | 87.5%                             | 0.67           | 78.77%                        |
> | 5        | 32                  | 93.75%                            | 0.56           | 78.43%                        |
> | 4        | 16                  | 96.88%                            | 0.44           | 77.40%                        |
> | 3        | 8                   | 98.44%                            | 0.33           | 70.31%                           |
>
> #  W2: Why the proposed sub-bit quantization method, i.e., OS-Quant, is effective.
>
> The effectiveness of our sub-bit quantization stems from two aspects.
>
> - **First is the data distribution-driven compression.** By analyzing weight distributions in well-trained BSNNs, we reveal obvious clustering patterns in convolutional kernels. This intrinsic characteristic enables compression of the full codebook for 3×3 kernels (512 combinations) into compact codebooks containing only high-frequency codewords (e.g., η=4 reduces to 16 codewords). This data distribution-driven compression preserves critical structural information during the quantization process.
>
> - **Second is the observation and resolution of the outlier-induced codeword selection bias in the vanilla sub-bit compression process.** Specifically, this sub-bit quantization method is highly sensitive to outliers. When a weight value in a kernel deviates significantly from the main distribution, it will lead to suboptimal codebook selection. Furthermore, our OS-Quant detects outlier boundaries through the interquartile range statistical method, and adaptively scales based on spatial neighborhood relationships. This not only eliminates the interference of outliers on the quantization process, but also fully preserves the spatial characteristics of the convolution kernel.
>
> Through the above two aspects, S²NN achieves compression below 1-bit while maintaining high model performance.
>
> # Q1 & W3: Construction of compact codebook.
>
> The compact codebook is established through initialization followed by learning. **During initialization**, a compact codebook $|ℙ| = 2^\eta$ is **randomly sampled from the full codebook** $\mathbb{K}$. *However, since random sampling cannot guarantee an optimal compact codebook, we refine the sampled subset through learning, following prior studies.* Specifically, $ℙ$ is represented as a tensor $\mathbf{p} \in \mathbb{R}^{k \times k \times |ℙ|}$ formed by concatenating initially sampled codewords. **During training**, $\mathbf{p}$ is **updated continuously**. Since $\mathbf{p}$ is no longer limited to {±1} during optimization, so sign(·) is applied on it after optimization to preserve the binary representation of the compact codebook $ℙ$.
>
> # Q2: Is the codebook generation sensitive to initialization or data distribution across datasets?
>
> We discuss the sensitivity of the codebook generation to initialization and data distribution.
>
> - Regarding the sensitivity of codebook generation to initialization. Although the codebook $ℙ$ is initialized through random sampling, gradient optimization during training ensures its final performance is insensitive to initial states. Stable convergence across multiple tasks corroborates this robustness.
>
> - Regarding the sensitivity of codebook generation to data distribution. For visual data, CNN and Transformer architectures exhibit strong clustering patterns, enabling consistent high compression efficacy. In contrast, non-visual data (e.g., SpikeLM for NLP) exhibit weaker clustering (Appendix A.8), resulting in about a 3% decrease in accuracy on GLUE tasks (73.7% vs. 76.5%, Table 10). Thus, the codebook generation is somewhat sensitive to data distribution.
>
> In summary, the codebook generation demonstrates initialization robustness but exhibits performance dependency on data modality (visual > non-visual).
>
> # Q3: The suitable value of η.
>
> η is a model-wise parameter rather than layer-wise, it represents the quantization bit-width. **Once we determine η, it applies to the entire model.** The selection of η should depend on the specific needs of the deployment:
>
> - **Larger η**: Optimal for accuracy-sensitive scenarios like safety-critical systems, preserving model performance with moderate compression.
> - **Lower η**: Suitable for resource-limited devices (e.g., mobile/IoT), maximizing compression and speed while accepting manageable accuracy trade-offs.
>
> # Q4: Analysis of the outlier detection threshold γ.
>
> Yes, we conduct an analysis by setting the outlier detection threshold γ to different values. Specifically, we test various γ values on the CIFAR100 dataset using the ResNet19 architecture, with η set to 6. The results are as follows:
>
> - γ=0.5 (too low) treats too many values as outliers, causing excessive outlier scaling and destroying the kernel's spatial information.
> - γ=3.0 (too high) fails to detect outliers effectively, degrading S²NN to baseline performance.
> - γ=1.5 achieves optimal balance between outlier detection and spatial preservation. This is the value we chose in the manuscript.
> - γ=1.0 and γ=2.0 perform well but slightly below γ=1.5.
>
> | γ    | 0.5   | 1.0   | 1.5   | 2.0   | 3.0   |
> | ---- | ----- | ----- | ----- | ----- | ----- |
> | Acc. | 74.48 | 75.34 | 75.59 | 75.19 | 75.08 |
>
> # Q5: Generalizability of outlier occurrence. Can the outlier be determined in advance?
>
> In our original manuscript, Fig 2(b) shows the first 60 kernels from ResNet19's second layer on CIFAR100, indicating the **prevalent issue of outlier occurrence** in a well-trained BSNN. **To further demonstrate that the emergence of outliers is a universal phenomenon, we explore whether the emergence of outliers varies by model, dataset, or data augmentation**. Specifically, we count the percentage of convolutional kernels containing outliers in each layer. Results are shown below, indicating that **while the number of outlier-containing kernels varies by model/dataset/augmentation, outliers occurrence remains a universal and severe issue**. Furthermore, as the network updates, the weights are continuously changed, resulting in varying outliers across training iterations. Consequently, the outliers cannot be determined in advance.
>
> |                                            | layer1.1  | layer2.1  | layer3.1  |
> | ------------------------------------------ | --------- | --------- | --------- |
> | CIFAR10, Res19                             | 21.15%    | 17.62%    | 22.33%    |
> | CIFAR100, Res19                            | 33.19%    | 18.99%    | 20.19%    |
> | CIFAR100, Res19, only RandomCrop           | 30.10%    | 26.24%    | 17.61%    |
> | CIFAR100, Res19, only RandomHorizontalFlip | 29.16%    | 21.61%    | 29.90%    |
> |                                            | **conv3** | **conv5** | **conv7** |
> | DVSCIFAR10, VGGSNN                         | 33.73%    | 15.99%    | 27.39%    |

---

### Official Review · Reviewer_TU7S · 2025-07-02

**Clarity:** 2
**Significance:** 2
**Originality:** 3
**Rating:** 3
**Confidence:** 4

**Summary:**

This paper proposes a sub-bit spiking neural network (S²NN) aimed at further exploring the compression and acceleration potential of spiking neural networks (SNNs). S²NN utilizes a compact codebook to encode weights into representations smaller than 1 bit, significantly reducing model storage and computational overhead. The authors introduce outlier-aware sub-bit weight quantization (OS-Quant), which optimizes codeword selection by identifying and adaptively scaling outliers.

**Questions:**

Does knowledge distillation in the method increase additional energy consumption?

**Ethical Concerns:**

["NO or VERY MINOR ethics concerns only"]

**Limitations:**

None.

**Paper Formatting Concerns:**

None.

**Quality:**

3

**Strengths And Weaknesses:**

Strengths: The paper features a comprehensive experimental design and a clear structure. By introducing sub-bit compression into SNNs, with weight representations below 1 bit (0.44 to 0.67 bit), it significantly enhances the potential for edge deployment. The paper features thorough experimentation, covering both vision and non-vision tasks, which validates the universal applicability of the proposed method.
Weaknesses: The paper lacks an in-depth theoretical analysis of OS-Quant and MPFD (e.g., convergence of gradient propagation, and why membrane potential-based distillation provides more precise optimization directions). In the classification task, more methods could be added for comparison.

---

> ### Author Rebuttal · Authors · 2025-07-31
>
> Dear reviewer TU7S, thank you for your review. Your concerns regarding our work primarily focus on four aspects: **(1) theoretical analysis of OS-Quant, (2) how MPFD provides a more accurate optimization direction, (3) more method comparison in the classification task, and (4) additional energy consumption for distillation**. We have provided detailed explanations for each of these concerns and hope to help resolve them.
>
>
> # W1: Theoretical analysis of OS-Quant about convergence of gradient propagations.
>
>
> - **Convergence of Vanilla Sub-bit Quantization.**
>   The vanilla sub-bit quantization converges using a straight-through estimator (STE) during backpropagation. In the forward pass, full-precision weights $\mathbf{w} _ {f,c}^{\ell}$ are mapped to the nearest binary codeword $\mathbf{k} \in \mathbb{P}^{\ell}$ via $\arg\min$, which is non-differentiable. During backward propagation, gradients $\partial\mathcal{L}/\partial\mathbf{w} _ {b,c}^{\ell}$ are approximated by bypassing the $\arg\min$ operation, treating it as an identity function within the STE. Formally, gradients are computed as $\partial\mathcal{L}/\partial\mathbf{w} _ {f,c}^{\ell} \approx \mathbb{1} _ {|\mathbf{w} _ {b,c}^{\ell}| \leq 1} \cdot \partial\mathcal{L}/\partial\mathbf{w} _ {b,c}^{\ell}$, ignoring quantization discontinuities. This allows gradient flow but introduces bias when outliers dominate the distance computation, leading to suboptimal codeword selection and unstable convergence.
>
> - **Convergence of OS-Quant.** OS-Quant ensures improved convergence by integrating outlier-aware adjustments into both forward and backward passes. During the forward pass, outliers in $\mathbf{w} _ {f,c}^{\ell}$ are detected via IQR and adaptively scaled using spatial neighbor relationships, yielding adjusted weights $\hat{\mathbf{w}} _ {f,c}^{\ell}$. Quantization then operates on $\hat{\mathbf{w}} _ {f,c}^{\ell}$. For backpropagation, gradients are computed via the chain rule: $\partial\mathcal{L}/\partial\mathbf{w} _ {f,c}^{\ell} = (\partial\mathcal{L}/\partial\hat{\mathbf{w}} _ {f,c}^{\ell}) \cdot (\partial\hat{\mathbf{w}} _ {f,c}^{\ell}/\partial\mathbf{w} _ {f,c}^{\ell})$. Compared with the vanilla sub-bit quantization, the term $\partial\hat{\mathbf{w}} _ {f,c}^{\ell}/\partial\mathbf{w} _ {f,c}^{\ell}$ explicitly accounts for outlier scaling, enabling gradients to reflect spatial corrections.
>
> - **Theoretical Advantage of OS-Quant.** OS-Quant converges better than vanilla quantization due to its outlier-robust gradient formulation. Let $\nabla _ {\text{vanilla}} = \partial\mathcal{L}/\partial\mathbf{w} _ {b,c}^{\ell}$ (simplified STE) and $\nabla _ {\text{OS-Quant}} = (\partial\mathcal{L}/\partial\hat{\mathbf{w}} _ {f,c}^{\ell}) \cdot (\partial\hat{\mathbf{w}} _ {f,c}^{\ell}/\partial\mathbf{w} _ {f,c}^{\ell})$. The scaling term $\partial\hat{\mathbf{w}} _ {f,c}^{\ell}/\partial\mathbf{w} _ {f,c}^{\ell}$ acts as a preconditioner: for outlier weights $(i,j) \in \mathbf{O} _ {f,c}^{\ell}, \partial\hat{\mathbf{w}} _ {f,c}^{\ell}(i,j)/\partial\mathbf{w} _ {f,c}^{\ell}(i,j) = 1/\Omega _ {i,j}$, where $\Omega _ {i,j}$ scales inversely with local weight variance. This downweights outlier gradients and amplifies spatially consistent signals. Consequently, $\nabla _ {\text{OS-Quant}}$ reduces gradient noise from outliers, while $\nabla _ {\text{vanilla}}$ propagates distorted gradients from biased codeword selections. The $\Omega _ {i,j}$-modulated gradients in OS-Quant thus promote stable descent toward minima that preserve kernel semantics, accelerating convergence and improving performance.
>
> # W2: Theoretical analysis of MPFD. (How does MPFD provide a more accurate optimization direction?)
>
> Dear Reviewer TU7S, we can understand that MPFD provides a more accurate optimization direction by analyzing the gradient of the distillation loss with respect to the membrane potential.
>
> Firing rate-based feature distillation (FRFD) methods treat the firing rate of neurons as network intermediate features and aim at aligning the firing rates between teacher and student networks. To achieve this alignment, the FRFD needs to adjust the membrane potentials in backpropagation and further control spike generation. Mathematically, the gradient of the distillation loss to the membrane potential is expressed as:
> $$
> \frac{\partial \mathcal{L _ {distill}}}{\partial\tilde{\mathbf{u}}^{\ell}[t]}=\frac{\partial\mathcal{L} _ {FRFD}}{\partial\mathbf{s}^{\ell}[t]}\cdot \frac{\partial \mathbf{s}^{\ell}[t]}{\partial g(\cdot)}\cdot\frac{\partial g(\cdot )}{\partial\tilde{\mathbf{u}}^{\ell}[t]},
> $$
> where $\frac{\partial\mathcal{L}_{FRFD}}{\partial\mathbf{s}^{\ell}[t]}$ can be directly calculated from the distillation loss function. Notably, **this gradient computation involves a surrogate gradient function $g(\cdot)$**, which causes the gradient induced by distillation on the membrane potential to be imprecise, thereby compromising the distillation optimization process.
>
> The proposed membrane potential-based distillation feature method (MPFD) applies distillation directly at the membrane potential level, enabling more precise optimization. Mathematically, **within MPFD, the gradient of the distillation loss to the membrane potential $\frac{\partial \mathcal{L_{distill}}}{\partial\tilde{\mathbf{u}}^{\ell}[t]}$ can be derived directly from our distillation loss function, resulting in less perturbation compared to FRFD method due to the absence of $g(\cdot)$ at this step.**
>
> # W3: More method comparison in the classification task.
>
> Thank you for your valuable suggestion. Due to space constraints, our manuscript compared only some advanced BSNN works. The revised version will add more comparisons of related works. Here, we **first present the added comparisons on the static ImageNet-1K and the neuromorphic DVS-CIFAR10.**
>
> ## ImageNet-1K
> On the ImageNet-1K, we compare related work in both ANN and SNN domains. As shown below, whether compared with the model compression methods in ANN or SNN, our S²NN consistently achieves close to SOTA performance under sub-1bit compression.
>
> > ||Method|Arc.|Bit (W/A)|Size (MBit)|OPs (G)|Acc. (%)|
> > |-|-|-|-|-|-|-|
> > | *Method in the ANN domain* | Bi-Real (IJCV 2020)| Res34            | 1/1        | -           | -        | 62.2      |
> > || IR-Net (CVPR 2020) | Res34  | 1/1        | -      | -        | 62.9      |
> > || SNN (ICCV 2021)  | Res34            | 0.67/1     | -           | -        | 61.4      |
> > || SNN (ICCV 2021)  | Res34            | 0.56/1     | -           | -        | 60.2      |
> > || SNN (ICCV 2021)           | Res34            | 0.44/1     | -           | -        | 58.6      |
> > || BiBert (ICLR 2022)        | Swin-T           | 1/1        | -           | -        | 68.3      |
> > || BinaryViT (CVPR 2023W)    | ViT              | 1/1        | -           | -        | 67.7      |
> > || ProxConnect++ (NIPS 2023) | ViT-B            | 1/1        | -           | -        | 66.3      |
> > || A&B (CVPR 2024)           | ReActA           | 1/1        | -           | -        | 66.9      |
> > | *Method in the SNN domain* | SDTv3 (TPAMI 2025)        | SDTv3-19M        | 32/1       | 607.57      | 16.03    | 79.80     |
> > || QSPIKF (CVPR 2024)        | SPIKFORMER-8-512 | 1/1        | 36.80       | 2.12     | 54.54     |
> > |  | AGMM (AAAI 2025)          | ResNet-18        | 1/1        | -           | -        | 64.67     |
> > || BESTF (IJCAI 2025)        | SPIKFORMER-8-512 | 1/1        | 44.56       | 5.67     | 63.46     |
> > || **S²NN**                  | **SDTv3-19M**    | **0.67/1** | **17.32**   | **0.84** | **68.02** |
> > | | **S²NN**                  | **SDTv3-19M**    | **0.56/1** | **15.88**   | **0.78** | **67.43** |
> > |  | **S²NN**                  | **SDTv3-19M**    | **0.44/1** | **14.31**   | **0.73** | **67.00** |
>
> ## DVSCIFAR-10
> On the DVSCIFAR-10, we add comparisons with more related BSNN works. As shown below, S²NN achieves near-identical accuracy to FP-SNNs (82.0% vs. 82.3% with TET), while reducing model size 38× (7.86 vs. 296.6 MBit) and operations 90% (0.2 vs. 1.97 OPs).
>
> > | Method                                  | Arc.       | Bit (W/A)  | Size (MBit) | OPs (G)  | Acc. (%) |
> > | --------------------------------------- | ---------- | ---------- | ----------- | -------- | -------- |
> > | TET (ICLR 2022)                         | VGGSNN     | 32/1       | 296.6       | 1.97     | 82.3     |
> > | ALBSNN (Frontiers in Neuroscience 2023) | 5Conv1FC   | 1/1        | -           | -        | 69.0     |
> > | CBP (IEEE JESTCS 2023)                  | 16Conv1FC  | 1/1        | -           | -        | 74.7     |
> > | Q-SNN (ACMMM 2024)                      | VGGSNN     | 1/1        | 10.91       | 0.31     | 81.6     |
> > | AGMM (AAAI 2025)                        | VGGSNN     | 1/1        | 10.91       | 0.31     | 82.4     |
> > | **S²NN**                                | **VGGSNN** | **0.67/1** | **7.86**    | **0.20** | **82.0** |
> > | **S²NN**                                | **VGGSNN** | **0.56/1** | **6.85**    | **0.17** | **81.6** |
> > | **S²NN**                                | **VGGSNN** | **0.44/1** | **5.74**    | **0.13** | **81.3** |
>
> # Q1: Additional energy consumption for distillation.
>
> We fully understand your concern about the additional overhead from knowledge distillation, but we would like to emphasize that it incurs zero energy consumption during deployment. **S²NN is designed to optimize SNN deployment efficiency while maintaining high performance.** By quantizing the weight to sub-bit, S²NN achieves extreme model compression. **Without distillation, S²NN will suffer from severe performance degradation and even convergence failures in challenging tasks.** Thus, distillation is essential for preserving S²NN's accuracy. Notably, distillation operates exclusively during training and remains fully decoupled from inference. **This ensures no additional inference overhead, aligning with our design goal for S²NN.**

---

### Official Review · Reviewer_d7VG · 2025-07-03

**Clarity:** 3
**Significance:** 3
**Originality:** 3
**Rating:** 4
**Confidence:** 4

**Summary:**

This work aims at addressing the storage and computational demands for large-scale spiking neural networks (SNNs). Its key idea is to employ sub-bit SNNs that represent weights with less than one bit (S2NNs). To do this, the work employs three main steps. First, it establishes an S2NN baseline that encodes weights using less than 1 bit. Second, it proposes an outlier-aware sub-bit weight quantization (OS-Quant) for improving binary kernel selection. Then third, it employs membrane potential-based feature distillation (MPFD) that aims to preserve the model’s performance.

**Questions:**

1)	Provide some evaluations for real-world deployment/scenarios, and if possible for bigger datasets as well.

2)	It mentions that “In the case of a 3×3 kernel, an analysis of the distribution of all possible 3×3 kernel values (23×3, representing the full codebook) reveals that only a small subset of binary kernels (i.e., codewords) is frequently activated”. This should be supported with some empirical data and/or illustrations.

3)	It also mentions that “… as networks scale up to meet practical application demands, the computational burden remains a challenge even with binary versions”. This should be supported with some empirical data to show that the existing binary versions are not enough.

4)	How to perform the sampling process to establish the compact codebook from the full codebook? It is important to clearly describe one of the main steps of the proposed techniques.

5)	The codeword selection process is somewhat unclear. Specifically, how the kernel of S2NN is obtained from its FP-SNN domain as shown in Figures 1-3.

6)	What variable “d” represents in Figures 1-3 is also not clear. It seems to be a critical part in understanding the proposed technique, especially for codeword selection.

7)	How to decide the suitable number of bits-per-kernel (η)?

8)	The sub-bit implementation is somewhat not clear. It is suggested to refine Figure 1(b) to illustrate the sub-bit implementation step-by-step while relating to examples in the description text. The discussion on the role of “d” may be critical here to help explaining the key steps. For this, information from appendix A.11 can be used in the main paper.

9)	It is also suggested to refine Figure 2 to show the steps in showing how actual outlier values from Figure 2(b) lead to outlier-induced codeword selection bias in Figure 2(a). Currently, Figure 2(a) and Figure 2(b) do not look connected to each other.

10)	What is the rationale of using the squared L2 criteria for identifying the outliers?

11)	This paper does not completely provide power and energy consumption results for all evaluated tasks (i.e., image classification, object detection, and semantic segmentation). Hence, it is recommended to complete these studies as well.

12)	There are several typos which need correction, such as “Nnormalized” in Figure 3(b) and “conclusin” in Ablation Study section.

**Ethical Concerns:**

["NO or VERY MINOR ethics concerns only"]

**Final Justification:**

Authors have addressed most of my comments, thought real-world evaluations can be improved. Anyway I think paper can be accepted, and authors have done a very good job in further improving this paper, and have sufficient feedback for future work as well.

**Limitations:**

Yes

**Quality:**

3

**Strengths And Weaknesses:**

Strengths

•	Its idea in applying sub-bit concept in SNNs is interesting and holds promising efficiency benefits.

•	The challenges of performance degradation in realizing sub-bit SNNs are addressed systematically.

•	It performs evaluation considering different datasets: image classification (i.e., CIFAR-10, CIFAR-100, ImageNet-1K, and DVSCIFAR-10), object detection (COCO), and semantic segmentation (ADE20K); and the results show promising improvements on model sizes, number of operations, and accuracy.


Weaknesses

•	The sampling process to establish the compact codebook from the full codebook is not sufficiently discussed.

•	The codeword selection process is somewhat still vague.

•	The study does not completely provide power and energy consumption results for all evaluated tasks (i.e., image classification, object detection, and semantic segmentation).

•	Evaluation for bigger datasets and real-world deployment/scenarios is missing

•	There are several issues with the technical descriptions and writing quality as highlighted in the question section below.

---

> ### Author Rebuttal · Authors · 2025-07-31
>
> Dear d7VG, thank you for your detailed feedback on our work. Below, we will address each of the weaknesses and questions you raised one by one. We hope this response can resolve your concerns.
> # Q1 & W4: Larger datasets and real-world evaluation.
>
> **The utilized datasets in our paper, ImageNet-1K, COCO, and ADE20K, are recognized as large-scale and challenging datasets in the fields of image classification, object detection, and semantic segmentation.** These datasets widely serve as standard benchmarks for evaluating various methods.
>
> **As per your suggestion, we have now included validations for real-world tasks, i.e., event-based object tracking tasks.** Experiments are conducted on three standard benchmarks, i.e., FE108, FELT, and VisEvent. We use a well-trained 0.56-bit S²NN on ImageNet-1k as the backbone for feature extraction, and other settings follow the advanced spike-based tracker, i.e., SDTrack[1]. As shown in the table below, with only 0.56-bit, our S²NN tracker surpasses most RGB-based trackers and is comparable to the state-of-the-art spike-based tracker SDTrack (all of which are full-precision). These results demonstrate the practical applicability of our approach in dynamic real-world environments.
>
> |Methods|Bit|FE108|FE108|VisEvent|VisEvent|
> |-|-|-|-|-|-|
> |||AUC(%)|PR(%)|AUC(%)|PR(%)|
> |STARK (ICCV 2021)|32|57.4|89.2|34.1|46.8|
> |SimTrack (ECCV 2022)|32|56.7|88.3|34.6|47.6|
> |OSTrack256 (ECCV 2022)|32|54.6|87.1|32.7|46.4|
> |ARTrack256 (CVPR 2023)|32|56.6|88.5|33.0|43.8|
> |SeqTrack-B (CVPR 2023)|32|53.5|85.5|28.6|43.3|
> |HIT-B (ICCV 2023)|32|55.9|88.5|34.6|47.6|
> |HIPTrack (CVPR 2024)|32|50.8|81.0|32.1|45.2|
> |ODTrack (AAAI 2024)|32|43.2|69.7|24.7|34.7|
> |STNet (CVPR 2022)|32|-| -|35.0|50.3|
> |SNNTrack (TIP 2025)|32|-| -| 35.4|50.4|
> |SDTrack$_{Tiny}$ (2025)|32| 55.3|88.1| 35.4| 49.5|
> |**S²NN**|0.56|54.3|85.5|35.4|49.5|
> >[1] SDTrack: A Baseline for Event-based Tracking via Spiking Neural Networks. 2025
>
> # Q2: Empirical data support for the clustering distribution of the binary kernel.
>
> We analyze **the total percentage of the top-k most frequent binary kernels** in binary ResNet and binary VGG, with results shown below. Clearly, by only using the most frequent 128 (from the total 512) binary kernels, the total percentage can surpass 74%. Besides, these clustering distributions become more noticeable in deeper layers.
>
> |Dataset|Top2|Top4|Top8|Top16|Top32|Top64|Top128|
> |-|-|-|-|-|-|-|-|
> |CIFAR100,Res19| 20.7% | 24.4% | 29.6% | 37.2% | 47.5% | 60.8% | 74.7%  |
> |layer1.2| 20.7% | 24.4% | 29.6% | 37.2% | 47.5% | 60.8% | 74.7% |
> |layer2.2| 21.8% | 24.8% | 30.0% | 38.6% | 50.2% | 65.0% | 80.0% |
> |layer3.1| 46.8% | 50.2% | 54.5% | 60.1% | 67.9% | 75.7% | 83.9% |
> |DVSCIFAR10,VGGSNN ||||||||
> | conv3| 15.1% | 18.1% | 23.6% | 31.8% | 43.2% | 59.1% | 74.7%|
> | conv5| 14.9% | 18.6% | 23.8% | 31.6% | 43.5% | 59.9% | 76.1%|
> | conv7| 14.2% | 18.0% | 24.6% | 34.1% | 47.7% | 65.8% | 82.3%|
>
> # Q3: Data support for 1-bit computational burden.
>
> Thanks for your valuable comments. Consider the spike-driven Transformer v3 (SDT-V3) [1]: even with 1-bit weight quantization, its best model requires ~21.63 MB storage and ~0.68 G operations. These resource demands may exceed the limits of low-resource hardware. For instance, the Loihi 1 neuromorphic chip has only 16 MB of neuron core memory[2,3], making it unable to deploy even the 1-bit compressed SDT-V3 model. In contrast, our method offers greater compression. With η = 4, the model size is reduced to about 9.5 MB and operations to 0.3 G, making deployment on resource-constrained hardware much more feasible.
>
> [1] Scaling spike-driven transformer with efficient spike firing approximation training. TPAMI 2025.
>
> [2] Loihi: A Neuromorphic Manycore Processor with On-Chip Learning. IEEE Micro 2018.
>
> [3] Loihi asynchronous neuromorphic research chip. Energy 2018.
>
> # Q4 & W1: Construction of compact codebook.
>
> We build the compact codebook in two steps: initialization and learning.
> -  **Initialization.** We randomly select a compact codebook $|ℙ| = 2^\eta$ from the full codebook to create the initial compact codebook.
> - **Learning.** Since random sampling may not be optimal, we refine ℙ through learning, following prior studies.* Specifically, $ℙ$ is stored as a tensor $\mathbf{p} \in \mathbb{R}^{k \times k \times |ℙ|}$ formed by concatenating initially sampled codewords. During training, $\mathbf{p}$ is updated continuously. Since $\mathbf{p}$ is no longer limited to {±1} during optimization, so sign(·) is applied on it after optimization to preserve the binary representation of the compact codebook $ℙ$.
>
> # Q5 & W2: Detailed codeword selection.
>
> We apologize for not clearly explaining the codeword selection and sub-bit implementation. Below, we outline the process.
>
> (1) **How is the kernel in S²NN baseline derived from the FP-SNN?**
>
> In the S²NN baseline, each FP kernel in the FP-SNN is replaced by the closest codeword from a compact codebook $\mathbb{P}$, following common sub-bit quantization approaches. Specifically, for a given FP kernel $w_f$, the codeword $k ∈ ℙ$ with the smallest L2 distance to $w_f$ is selected and used (namely *d* in  Fig 1-3)  for forward inference (Eq.5). A toy example is provided to illustrate this process:
>
> - FP kernel: $w_f$ = [0.8, 0.7, 5, -0.9, -0.8, -0.7, -0.9, -0.8, -0.9].
> - Two candidate codewords: $k_2$ = [1,1,-1,-1,-1,-1,-1,-1,-1], $k_3$ = [1,-1,1,1,1,1,1,1,-1].
> - Distances using L2 distance: *d*($w_f$, $k_2$) = 36.3, *d*($w_f$, $k_3$) = 35.5.
> - Codeword selection: Since *d*($w_f$, $k_2$) > *d*($w_f$, $k_3$), the S²NN baseline will use $k_3$ instead of $w_f$ for forward inference (Fig2a).
>
> (2) **Why does the baseline suffer from selection bias?**
>
> - **Ideal codeword**: Ideally, from a binarization view, $k_2$ best matches $w_f$, as it retains the sign patterns of the majority elements in $w_f$, except for an outlier value "5", while $k_3$ retains fewer correct signs.
> - **Selection bias**: However, the outlier "5" skews the distance metric, leading to incorrect selection of $k_3$. We define this inconsistency as the outlier-induced codeword selection bias.
> - **Prevalent outliers**: In Fig. 2b, we show the first 60 kernels of the second layer of ResNet19 on CIFAR100, showing prevalent outliers in FP kernels, which leads to severe sub-bit quantization error.
>
> (3) **How S²NN solve this problem?** (Fig.3)
>
> We introduce the OS-Quant to address the above bias, detailed as:
>
> - **IQR-based outlier detection** (Eq.7-8): For $w_f$=[0.8, 0.7, 5, -0.9, -0.8, -0.7, -0.9, -0.8, -0.9], so Q₁ = -0.9 and Q₃ = 0.75, thus the normal range of $w_f$ is [-3.375, 3.225]. This indicates that "5" in $w_f$ is an outlier.
> - **Spatially-aware scaling** (Eq.11-12): The outlier "5" is scaled to 1 using neighbor differences, yielding adjusted kernel: *ŵ* = [0.8, 0.7, 1, -0.9, -0.8, -0.7, -0.9, -0.8, -0.9].
> - **Outlier-Aware Sub-Bit Weight Quantization** (Eq.13): *d*(ŵ, $k_2$) = 4.3, *d*(ŵ, $k_3$) = 19.5. Therefore, S²NN correctly selects $k_2$, preserving more spatial sign patterns (Fig. 3a).
>
> # Q6: Clarification of *d*.
>
> We apologize for this missing. The variable *d* in Fig 1–3 is the squared L2 distance.
>
> # Q7: The suitable value of η.
>
> η is the bit-width used in quantization. **Once we determine η, it applies to the entire model, not just individual kernels.** The selection of η should depend on the specific needs of the deployment. Larger η is optimal for accuracy-sensitive scenarios, preserving model performance with moderate compression. Lower η is suitable for resource-limited devices, maximizing compression and speed while accepting manageable accuracy loss.
>
> # Q8: Step-by-step clarification of sub-bit implementation.
>
> Given a FP kernel $w_f$ and the compact codebook $ℙ={k_1,k_2,...,k_{2^\eta}}$, the vanilla sub-bit comprises three key steps:
>
> - **Step 1: Distance Computation**. For each codeword $k_i\in ℙ$, compute the distance between $w_f$ and $k_i$, i.e., *d*($w_f$,$k_i$)=$||k_i − w_f||^2_2$.
> - **Step 2: Nearest Codeword Selection.** This is done by comparing the distances calculated in Step 1. The selected codeword is $k_{\text{select}}=\arg\min_{k_i\in ℙ}d(w_f, k_i)$
> - **Step 3: Inference Execution.** Replace $w_f$ with $k_{\text{select}}$ for forward inference to complete the sub-bit quantization.
>
> # Q9: Relation between Fig 2(a) and Fig 2(b).
>
> Fig 2(a) is a specific example of Fig 2(b). Fig 2(a) aims to illustrate what the "outlier-induced codeword selection bias" is and how this issue affects the quantization process. Fig 2(b) presents the problem of prevalent outliers within the real FP-SNN.
>
> # Q10: Rationale for squared L2 criteria for identifying outliers.
>
> We draw on the advanced sub-bit quantization within the model compression domain and follow this calculation in our baseline. Notably, the novelty of OS-Quant lies in the IQR-based outlier identification and spatially-aware outlier scaling, rather than squared L2 criteria.
>
> # Q11 & W3: Lack of energy consumption results.
>
> We **adopt OPs instead of energy consumption** for efficiency evaluation **due to limitations in the existing energy calculation method.** Almost all SNN studies compute energy using 45nm technology with $E\_{MAC}$ = 4.6 pJ and $E\_{AC}$ = 0.9 pJ for FP32. However, our S²NN uses sub-1-bit beyond the first and final layer, making the FP32 $E\_{AC}$ value in the SNN inapplicable. After investigation, we find **no $E_{AC}$ value exists for binary operations under 45nm technology**. To ensure fair comparisons, we adopt OPs from model compression research [1,2]:
> $$
> OPs = FLOPs^{1} _ {32-bit} + \sum _ {l=2}^{L} SOPs _ {1-bit}^l, \text{ with } SOPs _ {1-bit} = fr \cdot T \cdot \frac{1}{64} FLOPs _ {32-bit}
> $$
>
> [1] BinaryBERT: Pushing the Limit of BERT Quantization. ACL 2021.
>
> [2] PokeBNN: A Binary Pursuit of Lightweight Accuracy. CVPR 2022.
>
> # Q12 & W5: Typos correction.
> Thank you for helping us improve the quality of our work. We will ensure that these corrections are made in our revised version.

---

> ### Comment · Reviewer_d7VG · 2025-08-05
> **Authors did an excellent job in addressing my comments**
>
> Overall, authors have done an excellent job in addressing my comments. Though evaluations in real-world settings are still not convincing, because real-world situations introduce so many uncertainties and environmental variations that typical dataset based evaluations fail. Anyway, authors may focus on this aspect in their future work.
> I will upgrade my rating to Borderline Accept.

---

> > ### Author Response · Authors · 2025-08-06
> > **Acknowledgment and Future Work Plan**
> >
> > Dear reviewer d7VG,
> >
> > Thank you for your thoughtful feedback. We are very pleased that our rebuttal has addressed your concerns. We will incorporate the related response into the revised manuscript to enhance its clarity and completeness.
> >
> > We agree that further investigation into real-world deployment is important. In future work, we plan to prioritize this by designing experiments that explicitly consider environmental variability, system noise, and other deployment challenges. We are also exploring collaborations to evaluate our approach in live or production-like settings.
> >
> > Thank you again for your constructive feedback and support.

---

### Note · Authors · 2025-08-14

We thank all reviewers for their valuable feedback, which greatly improved our manuscript. We appreciate reviewers’ recognition for our work, including: (1) **the first sub-bit SNN for enhanced edge deployment** (d7VG, WU5Y, LxTv, TU7S); (2) **identifying and addressing performance degradation caused by outliers** (d7VG); (3) **thorough experiments validating universal applicability** (d7VG, TU7S, LxTv, WU5Y); and (4) **hardware validation** (WU5Y, LxTv).

 During the review process, the main concerns raised by reviewers and our corresponding solutions are summarized as:

- **Construction of a compact codebook**: We provide a detailed construction process, including initialization and learning steps.
- **Additional method comparisons in the classification task**: We add comparisons on static ImageNet-1K and neuromorphic DVS-CIFAR10, covering both advanced ANN and SNN studies.
- **Additional energy consumption from distillation**: We clarify that distillation is used only during training. This ensures no inference overhead, aligning with S²NN's deployment goal.
- **Theoretical analysis of OS-Quant and MPFD**: We add a detailed theoretical analysis, including OS-Quant’s convergence in gradient propagation and MPFD’s role in providing a more accurate optimization direction.
- **Empirical data support for the clustering distribution and outlier occurrence**: We provide data statistics showing clustering patterns and confirming that outlier occurrence is a univeral and significant issue.

After the rebuttal phase, reviewers WU5Y and LxTv recommend acceptance, and reviewer d7VG praises our responses and suggests borderline accept. Although reviewer TU7S temporarily assigns borderline reject, we have provided detailed answers to all concerns and believe these fully address them. In summary, this paper present a sub-bit SNN that compresses weights below 1-bit, offering promising efficiency benefits. Our approach addresses codeword selection bias through OS-Quant and incorporates membrane potential-based feature distillation for enhanced performance. With SOTA results across various benchmarks and FPGA validation, we believe our S²NN makes a meaningful step to the SNN community, particularly for edge computing applications.

Once again, we extend our sincere gratitude to the AC and all reviewers for their time and thoughtful suggestions.

---

### Decision · Program_Chairs · 2025-09-17

**Decision:**

Accept (poster)

**Comment:**

The main idea of the paper is to use sub-bit SNNs that represent weights with less that one bit. First an encoding is presented based on clustering patterns of kernels. The authors mitigate an associated outlier induced codeword selection bias during training, by optimizing codeword selection. A membrane potential feature distillation procedure is also proposed.


Strengths identified by the reviewers include:

1. Interesting idea of using sub-bit concepts to SNNs, including the addressing of challenges associated with performance degradation. This should be useful for edge deployment.

2. Good evaluation across a diverse set of tasks (classification, object detection, segmentation, and NLP)

Weaknesses identified, which the authors should try to address, include:

1. The codebook generation process is a bit vague

2. Some writing quality issues

3. Lack of in-depth theoretical analysis of the algorithm

Overall the authors made a good effort addressing the concerns of the reviewers making some things in the exposition clearer vis-a-vis typos and evaluation, convincing some reviewers that the paper scores should be upgraded post rebuttal.

The paper received two Accepts, one Borderline accept and one Borderline reject. Its average ranking was above the typical acceptance threshold for NeurIPS, and in conjunction with the authors' addressing many of the reviewer concerns in the rebuttal leads me to recommend its acceptance.